METHODS

# A causal framework for the drivers of animal social network structure

Ben Kawam [1,2,3]*, Julia Ostner[1,2,4], Richard McElreath[3], Oliver Schülke[1,2,4☉], Daniel Redhead[3,5,6☉]

**1** Department of Behavioural Ecology, University of Göttingen, Göttingen, Germany, **2** Primate Social Evolution Group, German Primate Center Göttingen, Göttingen, Germany, **3** Department of Human Behaviour, Ecology and Culture, Max Planck Institute for Evolutionary Anthropology, Leipzig, Germany, **4** Leibniz ScienceCampus Primate Cognition, German Primate Center, Leibniz Institute for Primate Research, Göttingen, Germany, **5** Department of Sociology, University of Groningen, Groningen, The Netherlands, **6** Inter-University Center for Social Science Theory and Methodology, University of Groningen, Groningen, The Netherlands

☉ These authors contributed equally to this work.
* ben.kawam-1@biologie.uni-goettingen.de

**Data availability statement:** All relevant code and data can be found on our GitHub repository (https://github.com/BenKawam/causal_framework_ASN_structure).

## Abstract

A major goal of behavioural ecology is to explain how phenotypic and ecological factors shape the networks of social relationships that animals form with one another. This inferential task is notoriously challenging. The social networks of interest are generally not observed, but must be approximated from behavioural samples. Moreover, these data are highly dependent: the observed network edges correlate with one another, due to biological and sampling processes. Failing to account for the resulting uncertainty and biases can lead to dysfunctional statistical procedures, and thus to incorrect results. Here, we argue that these problems should be understood—and addressed—as problems of causal inference. For this purpose, we introduce a Bayesian causal modelling framework that explicitly defines the links between the target interaction network, its causes, and the data. We illustrate the mechanics of our framework with simulation studies and an empirical example. First, we encode causal effects of individual-, dyad-, and group-level features on social interactions using Directed Acyclic Graphs and Structural Causal Models. These quantities are the objects of inquiry, our *estimands*. Second, we develop *estimators* for these effects—namely, Bayesian multilevel extensions of the Social Relations Model. Third, we recover the structural parameters of interest, map statistical estimates to the underlying causal structures, and compute causal *estimates* from the joint posterior distribution. Throughout the manuscript, we develop models layer by layer, thereby illustrating an iterative workflow for causal inference in social networks. We conclude by summarising this workflow as a set of seven steps, and provide practical recommendations.

**Funding:** BK was supported by a grant from the German Research Foundation to OS as part of the Research Training Group 2070 "Understanding Social Relationships" (Project-ID 254142454). This research also benefitted from funds by the Deutsche Forschungsgemeinschaft (DFG, German Research Foundation) - Project-ID 454648639 - SFB 1528. DR was supported by the "Societal Transitions and Behavioural Change" sector plan from the ministry of Science and Culture of The Netherlands. The funders had no role in study design, data collection and analysis, decision to publish, or preparation of the manuscript.

## Author summary

Behavioural ecologists ask mechanistic questions about behaviour—causal questions. When studying animal societies, these questions often concern the drivers of social network structure. Addressing causal questions from observed social interactions, whether in wild or captive settings, poses serious inferential challenges. Social network data are often noisy and biased, and causal effects may be confounded. As a result, estimating the effects of interest requires careful causal and probabilistic modelling—tools that most empiricists in the field are not trained to use. By integrating techniques from causal inference and Bayesian statistics, we introduce a practical framework for researchers to conduct causal inference in their own study system. We start by distinguishing three levels of abstractions for any social network under scrutiny. We then outline an iterative workflow, built around a few key steps: (i) defining the causal effect of interest; (ii) translating one's domain expertise into qualitative, then quantitative causal assumptions; (iii) building a statistical model designed to estimate that effect. Throughout, we emphasise the justification and validation of statistical models, while offering guidance for readers who are unfamiliar with formal modelling. More broadly, our framework lays the groundwork for a stronger and more transparent bridge between theoretical and empirical research in behavioural ecology.

## Introduction

A major goal of behavioural ecology is to explain how ecological and evolutionary processes affect social structure [1]. Behavioural ecologists observe natural variation in social behaviour, and ask: "*why?*" [2,3]. Why do certain individuals cooperate by supporting each other against conspecifics, or by spending a substantial amount of time grooming? Why do other individuals confront each other in agonistic fights, sometimes at the risk of their lives? Ecology and evolution offer theoretical models to explain why individuals behave in certain ways [4–8].

Network Science provides valuable analytical tools to bridge theoretical predictions with empirical data [9]. To make inferences about the determinants of social structure, it is useful to first operationalise the system as a social network (see our Glossary in Table 1; [9–12]). Typically, a social network is composed of nodes that represent individual animals, and edges, which represent the social interactions or relationships between them. We find it important to distinguish three levels of abstraction for any social network under investigation (Fig 1). These levels differ regarding what the edges represent. In the highest level of abstraction, the network edges correspond to the *theoretical construct* that we most often wish to study (first level in Fig 1). That is the social relationship between two individuals, or an aspect of it like affiliation, dominance, agonistic support, tolerance, or friendship [7,13–16]. These constructs cannot be observed directly [6,17]. They are abstractions, often assumed to be composed of—or expressed as—a diverse range of behavioural interactions, and can only be approximated.

In the second level of abstraction, the network edges correspond to one type of quantifiable social interaction. These interactions are generally used as a proxy for the theoretical construct of interest [1,18]—though it can be the target network in some cases, e.g., when studying disease spread. Hereafter, we refer to this level as the *interaction network* (Fig 1), and assume that it is used as a proxy for a latent construct of interest. Researchers choose which behavioural proxy to use based on knowledge of their study system ("*How do I best approximate what I want to capture?*"). For instance, allogrooming (e.g., [19]), or spatial proximity

**Table 1. Glossary.**

| Term | Definition |
| --- | --- |
| Backdoor criterion | Graphical criterion providing a sufficient adjustment set (i.e. set of variables to include in a statistical model) for causal identification. Given a treatment $X$ and an outcome $Y$, a set of variable $Z$ satisfies the backdoor criterion if it does not include descendants of $X$ (i.e. nodes caused by $X$), and if it blocks all paths starting with an arrow pointing into $X$. |
| Bayesian model | Statistical model where inference is based on the posterior probability distribution, which describes the plausibility of different parameter values for all parameters in the model. The posterior probability is computed by combining the observed data with prior probability distributions for the model parameters, using Bayes Theorem. Bayesian models are also sometimes called *probabilistic models*. |
| Causal effect | A variable $X$ has a causal effect on a variable $Y$, if a (hypothetical) intervention on $X$ results in a change in $Y$: $P(Y) \neq P(Y\|do(X))$, in *do*-calculus. |
| Causal inference | Refers to both a discipline and a process. Discipline: field studying causal relationships among variables. Process: inferring causal effects from data. |
| Cercopithecinae | Sub-family of the African and Asian monkeys (sometimes referred to as "old world" monkeys) which comprises the baboons, the vervet monkeys, and the macaques. |
| Conditioning on a variable | Intuitively, conditioning on a variable amounts to saying "once we know its value". It is equivalent to "stratifying by" or "controlling for" it. In the context of a regression model, for example, adding covariates is a form of conditioning. |
| Confounder | A confounder of a "treatment" $X$ and "outcome" $Y$ is a variable that makes the observed statistical association between $X$ and $Y$ different than if $X$ had been intervened upon— for instance, using a randomised control experiment. That is, $P(Y\|do(X)) \neq P(Y\|X)$, in *do*-calculus. Note that confounding is a causal, and not a statistical, concept. |
| Directed Acyclic Graphs | Graphical causal model showing which variables are assumed to affect each other in a given study system. DAGs only encode qualitative knowledge. |
| Estimand | Target quantity for a given data analysis, defined outside of a statistical model. For instance, the total causal effect of a variable X on another variable Y in population Z. |
| Exogenous variable | Variables whose causes are not explicitly modelled. |
| Generative statistical model | Statistical model that can be used to simulate data, as they are implied by the model's assumptions. |
| Network structuring features | Variables shaping the network of social interaction. Features can be at the individual (e.g., age), dyadic (e.g., genetic relatedness), or supra-dyadic level (e.g., predation pressure). |
| Identifiability | A causal estimand is identifiable if it can be theoretically computed using observed data. For instance, suppose that $X$ causally affects $Y$, and that both $X$ and $Y$ are caused by an unobserved variable $U$. Then, the effect of $X$ on $Y$ is not identifiable, because of the confounder $U$. |
| Joint posterior probability distribution | A multidimensional probability distribution describing the plausible values of a statistical model's parameters, after it has been updated with data. |
| Licensing causal assumptions | Assumptions expressed in a formal language (e.g., Directed Acyclic Graph, mathematical equation) describing causal relationships among variables. These assumptions are *licensing* in that they provide conditions under which the causal interpretation of statistical estimates is justified from first principles. |
| Marginal posterior distribution | Posterior probability of a parameter regardless of (i.e. unconditional on) the value of the other parameters. |
| Markov Chain Monte Carlo | Algorithm to draw samples from—and thus, approximate—the joint posterior probability distribution of a Bayesian model. |
| Model precision | Refers to how variable posterior estimates are from one another across data samples (e.g., across iterations of a given SCM). More precise models are less variable across data samples. |
| Multilevel model | Also sometimes called "hierarchical" or "mixed-effect" models, multilevel models learn about the value of certain clusters using what they have learned in other clusters, when these clusters are part of a so-called "varying-effect" (or "random effect") structure. For instance, to estimate an individual $a$'s propensity to give social interactions to others, a multilevel model learns, naturally, from the interactions given from $a$ to others, but also from the other individuals' (e.g., $b$, $c$, $d$) propensities to give interactions to others. It does so through a process called *partial-pooling*. |

(*Continued*)

**Table 1**. (Continued)

| Term | Definition |
| --- | --- |
| **Open path** | On a DAG, a path is open if association flows through its components—i.e., two variables connected by an open path are statistically associated. On the other hand, a path is closed when the flow of association among its components is blocked. |
| **Outcome scale** | Scale of the outcome variable. For instance, rate of behavioural events, for our Poisson models. |
| **Regularised estimates** | Statistical estimates that tend to capture features of the target population (so called *regular* features), and not features of the specific sample (*irregular* features). |
| **Simulation study** | Small, synthetic world created—in our case—to understand how a statistical model behaves when confronted with a *known* structural causal model. |
| **Social network** | Graph where nodes represent individuals, and edges represent either (i) an aspect of the social relationship, (ii) the true interaction rates for a given behaviour, or (iii) the measured interaction rates for a given behaviour, between individuals. |
| **Social structure** | Pattern of social interactions and relationships among the members of a population. |
| **Statistical parameter** | Unobserved variable, whose value is estimated using a posterior probability distribution. Note that statistical parameters only describe associations, but contain no causal information. |
| **Structural Causal Model** | Type of causal model where the functional relations among variables can be specified with quantitative knowledge. For instance, certain causal effects (corresponding to arrows in the DAG) can be defined as strong, weak, linear, non-linear, etc. |
| **Structural parameters** | Parameters of a structural causal model. Note that structural parameters do contain causal information. |
| **Triadic closure** | In a triad composed of individuals $a$, $b$, and $c$, given that there is a connection between $a$–$b$, and one between $b$–$c$, there often exists a connection between $a$–$c$. |

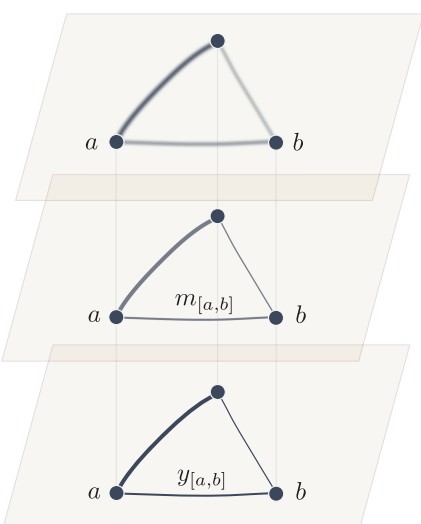

**1. CONSTRUCTS**
**Social relationship**
*Example:* affiliation
*Observed:* no

**2. INTERACTIONS**
**Behavioural proxy**
*Example:* true rate of grooming
*Observed:* no

**3. DATA**
**Sampled interactions**
*Example:* observed number of grooming bouts
*Observed:* yes

**Fig 1. Three levels of abstraction for an animal social network.** On the three levels, the dark dots, or *nodes*, represent individual animals, two of which are named $a$ and $b$. We show a network of only three individuals for simplicity of representation. The lines connecting individuals, or *edges*, depict the relationships or interactions among them. On all three levels, lines of different widths suggest variation in the strength of relationships, or in the number of interactions.

(e.g., [20]), may capture affiliative relationships in different species. Researchers also determine which proxy to use based on practical constraints ("*What can I measure?*"). Certain behavioural interactions may be reliable indicators of affiliation, but occur too rarely to be of practical use (e.g., "bridging" behaviour in macaques, see [21]). How the theoretical construct of interest relates to the interaction network is of critical importance for future research

(see [22–24]). It is, however, beyond the scope of the present manuscript. That is, we take for granted that the interaction network is a reasonable proxy for the theoretical construct of interest.

Third, it is typically impossible to observe every interaction event within the study system. Instead, researchers sample the study population using standardised behavioural protocols [25]. The data generated through the use of these protocols—e.g., 1% of all grooming bouts—are then used to approximate the network of all interactions [1]. These *data*, which we also further refer to as *sample* or *observations*, correspond to the third level of abstraction (Fig 1).

In the following sections, we use these concepts to describe common issues in animal social network analysis, and argue that they should be understood as problems of causal inference. We then introduce a novel, relatively simple, framework that explicitly addresses these issues.

## Common problems in animal social network analysis

Establishing the connection between observations (third level) and unobserved interaction network (second level) is a major inferential challenge in animal social network analysis [26]. A popular approach to approximate the interaction network is to *aggregate* the observed interactions at the individual- or dyad-level. These approaches include the use of Simple Ratio Index [27] and composite "Dyadic Sociality Index" [28], to quantify edge weight. The edge weights can be aggregated at the node level to calculate, for instance, node degree or strength [29]. These metrics are then commonly used as predictor or outcome variables in multivariable regression procedures [27].

However, using such network summary metrics in statistical models amounts to treating the unobserved interaction network (second level) as being observed. In practice, the observed interactions are *noisy samples* of the interaction networks [30,31]. Suppose we are interested in studying the rate at which two individuals groom. In one case, we observe these two individuals for 1 hour, and record 1 grooming bout (i.e. a grooming "event") between them. In a second case, we observe the same two individuals groom 100 times over 100 hours of observation. In both cases, the observed rate (i.e., the Simple Ratio Index) is equal to 1 grooming bout per hour. Yet, we are more *uncertain* about the real grooming rate in the first case, where the individuals have been observed for a shorter period. The data-aggregating procedures mentioned above would typically treat them as equivalent, and this invalidates probability calculations—different amounts of evidence must be reflected in inferential uncertainty. To deal with the uncertainty inherent to aggregating observations (hereafter referred to as *problem I*), behavioural ecologists traditionally attempt to collect a large amount of data to produce an accurate approximation of the unobserved interaction network [1,11,32]. But whether a given amount of data is "sufficient" for an accurate approximation of the network is hard to determine.

A related issue is that of "*biases*" introduced by the sampling process (*problem II*). The sampling regime itself may make some individuals appear more or less sociable than they actually are. Farine [33] provides an example where females are more gregarious than males in a hypothetical population of animals. However, females are sampled less regularly than the males (e.g., because the males are more visible). As a result, the two sexes appear equally gregarious. Permutation methods have been proposed to solve these sorts of issues, sometimes framed as problems of "non-independence of social network data" [27,33,34]. However, the utility of permutation methods for social network analysis has been challenged, as they carry several, fundamental, flaws [35–38]. A rich literature in network science [39–43], and recent work in animal social network analysis [23,24,31,38,44,45] proposes to solve the issues

of noise and biases introduced by the sampling process, by defining the true interaction rates as parameters in a Bayesian Model. We will return to this idea in detail, further below.

Social network data present yet another challenge—often framed as the "non independence in social network data" as well—but here, due to structuring features of the network. As opposed to the data dependencies caused to the sampling process, structural dependencies among edges are already present in the interaction network (second level). Consider, for instance, three edges from a hypothetical social network: the number of interactions given from an individual *a* to another individual *b*, the interactions from *a* to *c*; and those from *a* to *d*. The three edges will likely be more similar to each other than they are to the other edges in the network, because the actor, in the three cases, is *a* [46–48]. Interactions from *a* to *b*, and those in the opposite direction, from *b* to *a*, often covary as well. In the presence of dyadic reciprocity, dyads with a high rate of interaction in one direction will have a high interaction rate in the opposite direction too [49,50]. A third example regards triadic closure: it is often possible to learn about the value of the edge between *a–c*, given knowledge about the value of the edges *a–b* and *b–c*. This can be the case because *a*, *b*, and *c* all belong to the same kin group, where they engage in more interactions than with non-kin. The line between the two types of dependencies we have introduced (i.e. due to sampling, and structural) is however not strict. An individual's features might affect how they behave—creating structural dependency among its edges—and, in turn, its behaviour can affect the sampling process, e.g., if more gregarious individuals are sampled more regularly.

Ignoring structuring features of the network can have deleterious consequences, when the goal of a statistical analysis is to make inference about the effect of a given variable, like sex or age, on social behaviour. First of all, the effect of interest might be confounded (*problem III*). In this case, a statistical model describing the association between the predictor of interest and social behaviour will not recover the effect of interest, even with infinite sample size. Second, statistical models that ignore this structure can have low efficiency and low predictive accuracy (*problem IV*). That is, the models won't precisely recover generative parameters, and won't accurately predict unobserved data from the inferred interaction network [37,51]. How to build statistical models in light of structuring features of the network has received ample attention in the field of (human) Social Network Analysis [50,52–56]. Statistical models in behavioural ecology are rarely built in light of the structuring features of the social network, and might greatly benefit from incorporating these approaches.

In summary, inference from animal social networks implies several difficulties. Samples often look very different from the actual interaction network, because of noise (problem I) and potential biases (problem II) introduced by the sampling process. Structuring features of the social network further create associations between edges in the interaction network, and should be considered when building statistical models (problems III and IV). In the remaining sections of this manuscript, we argue that all of these issues are part of—and should be addressed as—one larger kind of problem: a problem of causal inference.

## Causal inference in animal behavioural ecology

Behavioural ecologists ask mechanistic questions about behaviour, causal questions. However, like other scientists, they face a dilemma. Many are taught that the *only* way to infer causality from data is to run a randomised controlled experiment [57]. Arguing otherwise can be perceived as endorsing the statement that "*correlation implies causation*" [58–60]. Yet, researchers often *cannot* run such experiments in their study system, for ethical or practical reasons. They must address causal questions using observational data, but under the idea that causal inference from non-experimental data is impossible.

As a result, many observational studies are causally ambiguous [61–63]. It is common to see scientific papers reporting observational studies with causal claims in their titles and abstracts—e.g., *X drives Y* in species *Z*. These papers begin with a largely causal reading of the literature (e.g., authors *J* found an *effect* of *X* on *Y* in species *W*). The Methods section, however, rarely contains transparent causal assumptions, even if statistical models are usually adjusted by "control" variables to avoid confounding biases—a clearly causal consideration. The Results section is most of the time free of causal language as well [59]. For instance, a variable, *X*, is said to be simply *associated*, or *correlated*, with the outcome of interest, *Y*. But the discussions and conclusions often turn to explicitly causal vocabulary again: the authors discuss causal explanations for the observed pattern of association, as well as their implications.

In the end, statistical models from which causal evidence is assessed have generally not been logically linked to licensing causal assumptions [64,65]. For this reason, it can be hard to evaluate how strong the causal evidence actually is. It may be difficult to know what exactly is the quantity a statistical model is supposed to estimate (what is the causal estimand), why is a statistical model by certain variables and not others (under which assumptions is the effect identifiable); or how to interpret the statistical parameters in terms of the causal structure of the study system [66–68].

Instead, inappropriate, non-causal approaches are often used to decide which "control" variables to include in a statistical model when the aim of the analysis is causal inference. A core problem is the use of predictive techniques that are given causal interpretation, for instance by selecting the predictor variables of a multivariable regression based on Information Criterion (e.g., AIC) or Cross-Validation procedures [69]. Another approach is to select covariates based on their p-values, for example by dropping non-significant variables. Researchers may also be advised to include all predictors that are assumed to affect the response variable. These approaches are *insufficient* to licence a statistical model designed to estimate a causal effect, because their logic has nothing to do with control of confounding, but other goals like forecasting or error control under random treatment assignment. In fact, well-meaning use of covariates may introduce biases—e.g., collider bias or posttreatment bias [37,58,60,63,67,70,71].

We propose an alternative to the approaches mentioned above, by integrating tools from the field of formal Causal Inference with Bayesian statistical modelling. Causal inference in experiments depends upon the logic of random treatment assignment for modifying a causal model of the system under study. Causal inference in observational settings depends upon the same logic, in the absence of random treatment assignment. Both settings, experiments and observation, require in principle an explicit model of the data-generating process, a causal model. The causal model, combined with one or more questions, can transparently and objectively justify one or more statistical models.

In our case, what is needed is a model of the causal factors driving the observed social interaction network, as well as a transparent workflow that links it to hypotheses and data analysis. We describe this workflow in three parts. First, we show how to represent hypotheses about social networks as formal causal models. For the sake of generality and communication, we use Directed Acyclic Graphs, or DAGs [60,65,67,70], to describe the causal structures underlying animal social interaction networks. Second, we show how to explore the empirical implications of the DAG by building Structural Causal Models (SCM), which can be used for synthetic data simulation. Third, we combine these two types of causal models with matched multilevel extensions of the *Social Relations Model*, a generative statistical model [23,37,38,47,50,72], and show that it can recover structural parameters from the simulations. This provides the grounds

for the development and justification of an empirical workflow in which statistical models have been developed and tested transparently under specific causal assumptions.

Our aim is not to provide a "one size fits all" model. Rather, the framework emphasises on the basic causal structure of animal social interaction networks, and on how they can be estimated with statistical models. Accordingly, empiricists interested in studying the drivers of social network structure can build upon our models, by tailoring them to their specific questions and study system.

## Mechanics of the framework

We build the framework step by step, through four simulation studies that incrementally present its elements in detail.

In simulation 1, we introduce the basic elements of our causal model. They define the link between the observed network (third level of abstraction) and the interaction network (second level), and specify assumptions about the factors shaping the interaction network. We encode—i.e. translate to a formal language—these assumptions using two types of causal models, a DAG and a SCM. We start by assuming that the structuring features are random. We then introduce the Social Relations Model, and show that it can recover the parameters of the SCM.

Next, we show how individual- (simulation 2) and dyad-level features (simulation 3) can affect the interaction network. We highlight that well-specified statistical models can accurately recover the causal effects encoded in the simulations. We also show how statistical models that are not adjusted by the structuring feature are affected by the causal effects, through their varying-effects structure, and discuss the interpretation of these effects. The aim of simulations 1-3 is not to be realistic. Instead, they reveal the internal *mechanics* of our framework.

In simulation 4, we turn to a more realistic scenario. We build an estimator for the effect of genetic relatedness on affiliation in females of a population of Assamese macaques, where this effect is confounded by dominance rank. We first validate our statistical model using synthetic data, and then fit it to empirical observations. Throughout the simulations, we show how our framework naturally addresses the issues mentioned above (problems I to IV).

### Simulation study 1: Random structuring features

**Introduction to directed acyclic graphs.** DAGs are a simple and powerful tool to describe how variables in a system causally affect one another [67,70]. The mathematical and applied literature on DAGs is large. We give here only a necessary conceptual introduction. DAGs are simple, but they are sufficient to describe the fundamental obstacles to, and derive solutions for, causal inference—including the design of experiments and observational studies.

DAGs are *graphs*. They are composed of nodes—here, representing variables—and directed edges, encoding causal relationships among these variables. For instance, the following DAG:

$$X \to Y$$

can be read as: "$X$ influences $Y$". There is no universal definition of what "influences" means in science, but in a DAG it means that an intervention on $X$ changes the distribution of $Y$. In contrast, an intervention on $Y$ would not change $X$, because of the direction of the arrow. It is important to note that DAGs are qualitative: they do not say anything about the functional

form of the causal effects. They simply posit the presence and absence of causal relationships among a set of variables.

A *path* is a sequence of two or more adjacent nodes (i.e., nodes directly connected by an arrow). For example, consider:

$$X \rightarrow Z \rightarrow Y.$$

Here, $X$ transmits a causal effect to $Y$, through a *mediator*, $Z$. This specific structure (i.e., a path involving three nodes connected by two arrows pointing in the same direction) is referred to as a *chain*. Chains are a type of *causal* paths, as the arrows connecting its two ending points—here, $X$ and $Y$—go in the same direction. Paths can also be *non-causal*, as in:

$$X \leftarrow Z \rightarrow Y.$$

This particular path, where a variable $Z$ is a common cause of two other variables $X$ and $Y$, is called a *fork*. It is non-causal because one of the arrows between $X$ and $Y$ goes "backwards".

Although causal effects (consequences of interventions) only flow through causal paths, statistical association can arise through non-causal paths too. This is a fundamental reason that statistical association does not imply causal relationships. For instance, $X$ and $Y$ are associated in a fork, because of their common cause $Z$. Their association arises from a common cause, not the influence of one on the other.

A graph is *acyclic* if it is impossible to start at any node, and return to it by following causal paths only, or in other words: a node never influences itself. If one were to model feedback loops, they would need to represent time explicitly on a DAG (e.g., $X_t \rightarrow Y_t \rightarrow X_{t+1}$; see [73]). What makes DAGs *directed* is that causal arrows can only be single-headed. Double-headed arrows can however be used as shortcuts, to indicate that an unspecified non-causal path connects two variables. For instance, if an unobserved variable $U$ affects both $X$ and $Y$, it can be represented as:

$$X \longleftrightarrow Y, \text{ which is equivalent to: } X \leftarrow U \rightarrow Y.$$

For a general introduction to causal graphs, see Pearl [67] and McElreath [37].

**Directed acyclic graph for simulation study 1.** In Table 2, we introduce the core elements of our causal framework. It links observed social interactions $y$ in a dyad to the unobserved true rate of interaction $m$ in that dyad, which is caused by features of the individuals, $\gamma$ and $\rho$, and of the dyad $\tau$. Each individual is assigned a general tendency to perform a particular behaviour, across all dyads. We call this *giving*, and represent it with the parameter $\gamma$ ("gamma"). Each individual is also assigned a general tendency to receive a behaviour, across dyads. We call this parameter *receiving*, and represents it with $\rho$ ("rho"). In addition to these general tendencies to give and receive, each individual can have a specific *tie* with each other individual. We denote these ties with $\tau$ ("tau"). Ties, like $m$ and $y$ are directional—their values in the two directions (from $a$ to $b$, and from $b$ to $a$) are not necessarily the same.

We illustrate how these parameters affect one another in Fig 2A. This DAG embodies assumptions about *which factors shape each observed edge* $y_{[a,b]}$. It is the simplest data-generating process in our framework: a system where $\gamma$, $\rho$, and $\tau$ share no common influences. More formally, we can say that they are only affected by exogenous, independent variables. The most common way to represent this scenario on a DAG is to draw no arrow pointing into these variables. Thus, the propensity of an individual, $a$, to give interactions to others ($\gamma_{[a]}$), its propensity to receive interactions from others ($\rho_{[a]}$), and the propensity it has

**Table 2**. **Core elements of the causal framework.** Variables encode assumptions about individual- and dyad-level features on social interactions. See section I in S1 Text for group-level effects. The parameters $G$, $R$, and $T$ differ across models and are not defined here.

| Parameter | Indexed by | Definition/Function | Example | Notes |
|---|---|---|---|---|
| $y$ | Directed dyad, i.e., one value per dyad, and per direction: $y_{[a,b]}$ (direction 1) $y_{[b,a]}$ (direction 2) | Number of *observed* social interactions given from an individual, $a$, to another individual, $b$, over a sampling period. | N. of observed grooming bouts from $a$ to $b$ over 12 hours of combined focal follows of $a$ or $b$. | $y$ has the same meaning as in Fig 1, except that here, the interactions are directed. |
| $m$ | Directed dyad: $m_{[a,b]}$ (direction 1) $m_{[b,a]}$ (direction 2) | True rate/number of social interactions given from an individual, $a$, to another individual, $b$, over a sampling period. | Average number of grooming bouts given by $a$ to $b$ per period of 12 hours. | $m$ is unobserved (see Fig 1-2). It has the same meaning as in Fig 1, except that here, the interactions are directed. |
| $\gamma$ (*giving*) | Individual: $\gamma_{[a]}$ | Encodes assumptions about the effect of individual-level features on the general tendency of an individual to *give* social interactions to others in the group. | Given that $X_{[a]}$ represents the age of an individual $a$, $X_{[a]} \rightarrow \gamma_{[a]}$ can be read as: "*age affects how many social interactions an individual, a, tends to give to others*". | $\gamma$ can, for example, encode the assumption that older individuals spend less time engaging in social interactions (e.g., grooming), because they may be constrained by their low foraging efficiency. |
| $\rho$ (*receiving*) | Individual: $\rho_{[a]}$ | Encodes assumptions about the effect of individual-level features on the general tendency of an individual to *receive* social interactions from others in the group. | Given that $X_{[a]}$ represents the age of an individual $a$, $X_{[a]} \rightarrow \rho_{[a]}$ can be read as: "*age affects how many social interactions an individual, a, tends to receive from others*". | $\rho$ may, for instance, describe that individuals of a certain age are more attractive interaction partners. |
| $\tau$ (*tie*) | Directed dyad: $\tau_{[a,b]}$ (direction 1) $\tau_{[b,a]}$ (direction 2) | Encodes assumptions about the effect of dyad-level features on the tendency of an individual, $a$, to give social interactions with a specific other individual, $b$. | Given that $X_{|a,b|}$ is the genetic relatedness between $a$ and $b$, $X_{|a,b|} \rightarrow \tau_{[a,b]}$ encodes the following assumption: "The relatedness between $a$ and $b$ affects how many interactions $a$ gives to $b$". | We index *symmetric* dyadic variables, like genetic relatedness, using vertical bars ($X_{|a,b|} = X_{|b,a|}$). We index *directed* dyadic variables using square brackets ($\tau_{[a,b]}$ is not necessarily equal to $\tau_{[b,a]}$). |

to give interactions specifically to any other individual, $b$, in its group ($\tau_{[a,b]}$), are all independent. These three parameters affect the true rate of interactions from $a$ to $b$: $m_{[a,b]}$. Then, $m_{[a,b]}$ affects the observed number of interactions, $y_{[a,b]}$. This slightly contrasts with Fig 1, where the link between $y$ and $m$ was not explicitly causal.

We now have a graphical representation of our causal model for simulation 1. However, what it exactly means for a variable to affect the "tendency" of an individual to give, or receive, interactions, as well as the exact nature of the above-mentioned "exogenous variation", depends on the mathematical function that we assign to it. This is why, in the next section, we turn the qualitative assumptions of the DAG into quantitative assumptions.

**Structural causal model.** Structural Causal Models (SCM) are a type of causal model where the functional relationship among variables is defined by a mathematical function [70]. It means that they contain assumptions about whether the effects among variables—corresponding to arrows on a causal graph—are positive, negative, strong, linear, etc. SCMs are closely linked to causal graphs, such that there exists a causal graph for each SCM. As SCMs encode quantitative causal assumptions, they can be used as *generative models*, to simulate observations.

Practically, we used SCMs for data simulations that "obey" the causal relations represented on the DAGs, and implemented them in R, version 4.2.1 [74]. For each arrow on the DAGs, we defined a function, with arbitrary parameter values. We did so with two main goals. First, each SCM serves as a simple *example* of data-generating process encoded by its DAG. These simulations can be more tangible than the abstract, non-parametric, knowledge encoded in causal graphs. Second, and perhaps most importantly, SCMs can be used to derive and validate *statistical models*. We will focus on the former goal in this section, before turning to the latter, in the next section.

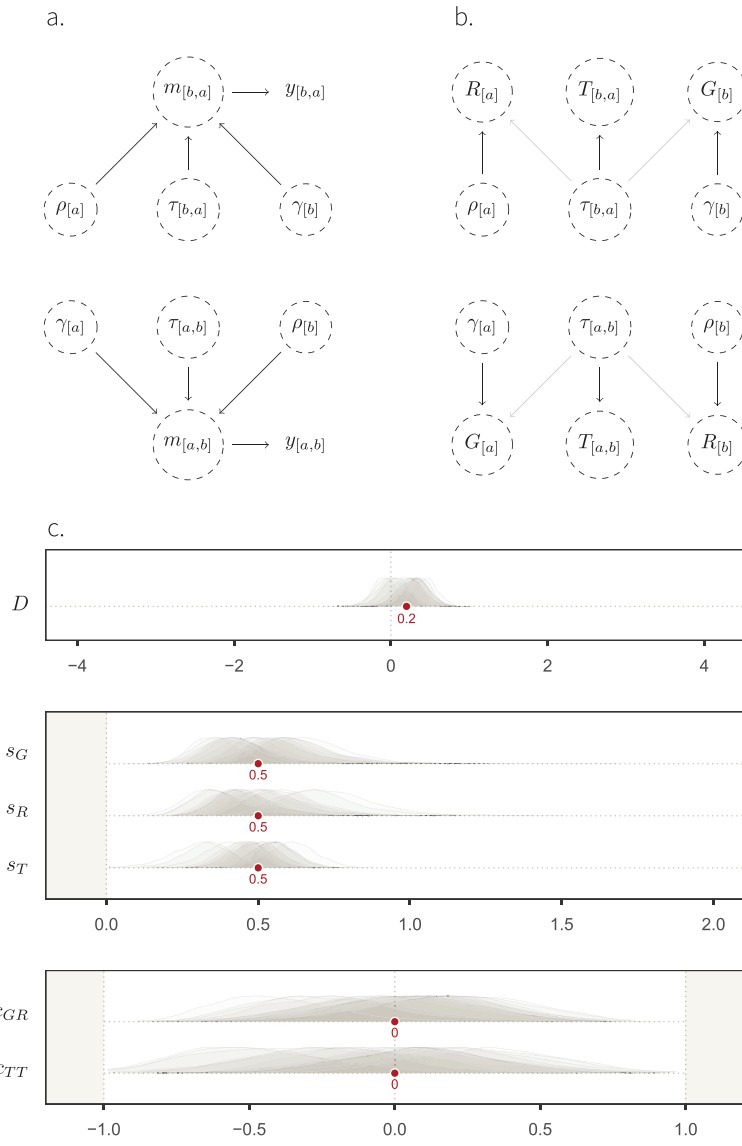

**Fig 2. Simulation study 1. A.** DAG showing the core elements of our causal framework, and how they causally affect one another, for two individuals, $a$ and $b$. No arrows enter the structuring parameters $\gamma$, $\rho$ and $\tau$. This means that exogenous, independent, noise is the only factor affecting: (i) how many social interactions an individual, $a$, generally gives to others ($\gamma_{[a]}$); (ii) how many interactions $a$ generally receives from others ($\rho_{[a]}$); and (iii) how many interactions $a$ specifically gives to another individual, $b$ ($\tau_{[a,b]}$). These three structural parameters affect the true rate of interactions given from $a$ to $b$ ($m_{[a,b]}$), which causes the number of observed interactions ($y_{[a,b]}$). We show the two directions—i.e. from $a$ to $b$, and from $b$ to $a$, and represent unobserved variables with a dashed circle. **B.** Mapping between structural parameters (Greek letters) and statistical parameters (Latin letters) of the non-adjusted Social Relations Model (Eqs 1.2.1–1.2.4). The transparent arrows between the structural dyadic parameter $\tau$ and the statistical individual parameters $G$ and $R$ indicate that these effects are possible (e.g. simulation 3), but do not exist here. **C.** Marginal posterior distributions, over a range of parameter values (*x-axis*) for the fixed-effects (*y-axis*) of posterior model 1. Their respective target values are shown as red dots. These fixed effects—except for $D$ which does not appear on the DAG—capture the patterns of (co-)variation of $G$, $R$, and $T$, shown in panel **b**.

Below, we provide a mathematical description of the SCM for simulation study 1—hereafter *SCM 1*. We start by defining $f_\gamma$, the function determining $\gamma$ for each individual:

The function determining gamma is defined as follows: ...is drawn from the following distribution:

$$f_\gamma: \quad \gamma_{[a]} \sim \mathrm{Normal}(0, 0.5) \tag{1.1.1}$$

The value of gamma, for each individual, $a$... Mean SD

Where $a \in \{1, ..., N\}$. This notation with the $\in$ symbol implies that $a$ represents an arbitrary individual index that can take any value from 1 to $N$, where $N$ is the number of unique individuals in the sample. Accordingly, Eq 1.1.1 should be interpreted as follows: $\gamma_{[1]}, \gamma_{[2]}, ..., \gamma_{[N]}$ are distributed as a normal (or Gaussian) distribution with mean 0 and standard deviation 0.5.

$$f_\rho: \quad \rho_{[a]} \sim \mathrm{Normal}(0, 0.5) \tag{1.1.2}$$
$$f_\tau: \quad \tau_{[a,b]} \sim \mathrm{Normal}(0, 0.5). \tag{1.1.3}$$

$b$, like $a$, is an arbitrary individual index (where $b \in \{1, ..., N\}$ and $a \neq b$). Thus, $\tau_{[a,b]}$ represents the value of $\tau$ for any combination of $a$ and $b$:

$$\tau_{[a,b]} \in \{\tau_{[1,2]}, \tau_{[2,1]}, ..., \tau_{[N-1,N]}, \tau_{[N,N-1]}\}.$$

In Eqs 1.1.1–1.1.3, we have defined what it meant for $\gamma$, $\rho$, and $\tau$ to be only affected by exogenous, independent, variables. We can now turn to define how these structural parameters affect $m$, the true interaction rate:

$$f_m: \quad m_{[a,b]} = \exp(0.2 + \gamma_{[a]} + \rho_{[b]} + \tau_{[a,b]}). \tag{1.1.4}$$

The intercept value 0.2 is the logarithm of how many interactions individuals give to, and receive from, others in the network for an average dyad. Here, this average rate is $\exp(0.2) \simeq 1.2$ interactions. The exponential function of $f_m$, in combination with the additive (or "linear") combination in terms, ensures that the interaction rate $m$ is always positive. It also has the following consequence: the effects of $\gamma$, $\rho$ and $\tau$ on $m$ are now *multiplicative*. It means, for instance, that the effect of $\tau_{[a,b]}$ on $m_{[a,b]}$ is larger if $\gamma_{[a]}$ and $\rho_{[b]}$ have large values themselves. We will come back to this issue, because it is a fundamental and unavoidable aspect of modelling behavioural count data with a Generalised Linear Model: causal effects are not simple and additive on the outcome scale, even though we model them as additive on the link scale.

The rate of behaviour $m_{[a,b]}$ is connected to the observable count $y_{[a,b]}$ through a stochastic process. The simplest choice is a Poisson process:

$$f_y: \quad y_{[a,b]} \sim \mathrm{Poisson}(m_{[a,b]}). \tag{1.1.5}$$

Eq 1.1.5 defines the relationship between the true interaction rate (second level of abstraction) and the observed number of interactions (third level), represented in Fig 1. The number of observed interactions $y_{[a,b]}$ is drawn from a Poisson distribution, characterised by one parameter: its average rate of interaction, $m_{[a,b]}$, specific to each directed dyad.

In summary, SCM 1 mathematically defines *one* simple data-generating process compatible with the DAG in Fig 2A. The generated data is represented in Fig A of S1 Text. Note that the number of individuals in the network, the value of the intercept in $f_m$, or the distributions we have chosen (normal, Poisson), are flexible. We might have, for instance, decided to model binary social events (e.g., has *a* been seen grooming *b* during a focal-animal sampling protocol?) instead of unbounded counts, which would make a Bernoulli distribution rather than Poisson appropriate. The key idea in any case is that the target of inference, the rate *m* and its components, is not observed but must be modelled as filtered by a stochastic sampling process. This stands in contrast to aggregated indices of sociality (e.g., Simple Ratio Index) which produce point estimates of rates, without properly characterizing uncertainty, before decomposing them as functions of explanatory variables.

In the next section, using the simple simulation defined above, we build a statistical model that aims to *uncover*—or, in this case, because we are dealing with synthetic data, *recover*—the parameters of the data-generating process.

**The social relations model.**   A non-parametric causal model, like a DAG, defines a qualitative hypothesis about how observations arise. A SCM makes this hypothesis into a data-generating algorithm that can be used for planning, understanding, and validating projects. Using data, whether simulated or real, requires a statistical model. Here, we explain how a statistical network model relates to a DAG and SCM, and how it can be developed and validated together with the causal models.

Statistical models and causal models are similar but distinct. Causal models contain directionality—they define the consequences of interventions—while statistical models do not. Statistical models contain additional elements that are useful for estimation. Importantly, they can include hierarchical distributions that function to estimates. These elements exist in both Bayesian and non-Bayesian approaches, because they produce better, more produce better, more efficient estimates [51].

The statistical models that we use in our framework are extensions of the "*Social Relations Model*", initially developed in the social sciences by Kenny & La Voie [47]. Compared to the initial model, ours have a multilevel varying-effect structure, can include covariates [72], and have non-Gaussian likelihoods [50]. They are further implemented in a Bayesian framework (see [37]; chapter 14) and can incorporate flexible additive effects, such as a stochastic block-model, i.e. a model capable of estimating causal effects of observable categorical variables, like sex, on network structure [23,38]. We wrote the models presented in this manuscript in the *Stan* probabilistic programming language [75], and ran them through *R*, using *CmdStanR* [76]—note however that the Social Relations Model can be implemented using R packages like *Rethinking* [37], *STRAND* [23,38], or *BISoN*[31].

In practice, we use statistical models as *estimators*. Given (i) an *estimand*, and causal assumptions encoded in (ii) a DAG and (iii) a SCM, we build a statistical model whose goal is to *estimate* the estimand. In this section, we outline how such a model can learn, by fitting it to known data: the data generated through the simulation above (SCM 1). We feed the statistical model with the number of observed interactions $y_{[a,b]}$ for each directed dyad in the network. Given these data, the model has to recover the unobserved rates $m_{[a,b]}$, as well as other structural parameters. In this first simulation, we do not focus on one particular estimand, but instead describe how the statistical model generally learns about the social network structure.

Below, we mathematically define the statistical model of simulation study 1 (hereafter statistical model 1). We start by describing the observed data *y*—here, one data point per directed dyad—using a conservative choice of probability distribution for unbounded count

data: the Poisson distribution [37,77].

$$y_{[a,b]} \sim \text{Poisson}(m_{[a,b]}) \tag{1.2.1}$$

This line can be read as such: "*$y_{[a,b]}$ is described using a Poisson distribution, whose average rate is $m_{[a,b]}$*". Here, $y_{[a,b]}$ is known, and $m_{[a,b]}$ is not: it has to be estimated. Recall that as opposed to Eq 1.1.5, we are, here, describing a *statistical* model, which *learns* the value of unobserved variables, like *m*. For this reason, the model will return a posterior probability distribution for each parameter $m_{[a,b]}$. Statistical models and SCMs can be visually differentiated in this manuscript, because each structural equation is defined by a function *f*. $m_{[a,b]}$ is further defined in terms of several parameters:

$$m_{[a,b]} = \exp(D + G_{[a]} + R_{[b]} + T_{[a,b]}) \tag{1.2.2}$$

As in the SCM, the exp (exponential) function ensures that *m* remains positive; it corresponds to the so-called inverse-link function. *D*, *G*, *R*, and *T* are all unobserved. Like *m*, they have to be estimated, and will be assigned a posterior probability distribution.

*D*, the intercept, is fixed across all dyads. It is equivalent to the intercept of 0.2 in SCM 1, which it should recover. In this simulation study, *G*, *R*, and *T* can be thought of as the statistical counterpart of the structural parameters $\gamma$, $\rho$, and $\tau$ (Fig 2B). $G_{[a]}$ captures the general propensity (i.e. the average rate, on the link scale), of an individual, *a*, to give social interactions to other individuals in the network. It is only affected by $\gamma_{[a]}$, and hence, $G_{[1]}$ should, for instance, recover $\gamma_{[1]}$. $R_{[b]}$, on the other hand, captures the propensity of an individual, *b*, to generally receive social interactions from others. It is, analogously, affected by $\rho_{[b]}$ only. Finally, $T_{[a,b]}$ represents the residual tendency of *a* to give interactions to *b*, conditional on $G_{[a]}$ and $R_{[b]}$. Here, it should recover the dyad-specific tendency effect $\tau_{[a,b]}$. In later simulation studies (e.g., in simulation study 3), we will see cases where the statistical and structural parameters are not equivalent.

To efficiently estimate the components of the rate $m_{[a,b]}$, it is useful to employ hierarchical distributions that relate the components to one another and allow them to share information through those statistical relationships. In practice, this means that $G_{[a]}$ and $R_{[a]}$ are described as varying-effects in a Multivariate Normal distribution.

The average individual giving and receiving rates...    Variance in the dist. of G    Covariance between G-R

$$\begin{pmatrix} G_{[a]} \\ R_{[a]} \end{pmatrix} \sim \underline{\text{MVNormal}} \left[ \begin{pmatrix} 0 \\ 0 \end{pmatrix}, \begin{pmatrix} s_G^2 & c_{GR}s_G s_R \\ c_{GR}s_G s_R & s_R^2 \end{pmatrix} \right] \tag{1.2.3}$$

...are described using a multivariate normal dist., whose parameters are:    Mean    Variance in the dist. of R

The distribution, in this case, has two dimensions: *G* and *R*. Its mean of $\{0, 0\}$ implies that $G_{[a]}$ and $R_{[a]}$ are *deviations* from the average interaction rate in the network, *D*. Centring the distribution on $\{0, 0\}$ also ensures that *G* and *R* can be uniquely estimated from the data. $s_G$ and $s_R$ describe the amount of variation in $G_{[a]}$ and $R_{[a]}$, respectively. If, for instance, individuals vary a lot in how many interactions they give to others, but do not vary much in how many interactions they receive from others, then $s_G$ will be large, and $s_R$ will be small. The pattern of covariation between these two dimensions is then captured by $c_{GR}$. Suppose a network

where individuals who give a lot of interactions also receive a lot of interactions. This situation would result in a positive estimate for $c_{GR}$.

Similarly the directed ties in the network are modeled hierarchically, using parameters for structure within and among dyads:

$$\begin{pmatrix} T_{[a,b]} \\ T_{[b,a]} \end{pmatrix} \sim \text{MVNormal}\left[\begin{pmatrix} 0 \\ 0 \end{pmatrix}, \begin{pmatrix} s_T^2 & c_{TT}s_T^2 \\ c_{TT}s_T^2 & s_T^2 \end{pmatrix}\right]. \tag{1.2.4}$$

Eq 1.2.4 is similar to Eq 1.2.3, with one exception. There is only one variance parameter in the variance-covariance matrix: $s_T^2$. Recall that $a$ and $b$ are only arbitrary labels. Thus, there is no reason why the variation in $T_{[a,b]}$ should be different from that of $T_{[b,a]}$. $c_{TT}$ hence describes the association between the number of interactions that $a$ gives to $b$, and those that $b$ gives to $a$, conditional on their respective averages $G_{[a]}$ and $R_{[b]}$. We will see further below how this parameter may be interpreted biologically, in light of causal models. We provide details about the prior probability distributions, as well as the exact parametrisation of our statistical model, in the sections A.2–A.3 of S1 Text.

In this section, we have defined the basic architecture of the Social Relations Model (which we sometimes refer to as *non-adjusted* Social Relations Model). A feature of this model we want to highlight is that it describes the true rate of interactions $m$ as a *statistical parameter*, for which a probability distribution is computed using Bayesian updating. This represents a principled solution to deal with the uncertainty and noise inherent to sampling social interactions (problem I). It notably contrasts with data-aggregation methods that we mentioned above, which treat data $y$ (third level of abstraction) as the *known*, true interaction rates (second level). Moreover, the varying effects of the Social Relations Model explicitly describe several patterns of (co)variations—i.e. structural "dependencies"— present in social networks, like those mentioned in the introduction (problem IV). This results in more accurate and efficient estimators [37,51].

**Posterior model.** In this section, we describe *posterior model 1*: the model fit obtained by feeding the data generated by SCM 1 into statistical model 1. We ran 15 iterations of the SCM, and fitted the data from each iteration into the statistical model. Thus, we obtained 15 marginal posterior distributions per "fixed-effect", or population level parameters (Fig 2C). We show each parameter's posterior distributions with its *target value*, i.e. the value it should recover, given that we know the generative model. The pattern one would expect from an estimator that accurately recovers structural parameters is the following: over several iterations, the high-density regions of the marginal distributions should overlap with the target value. Accordingly, we see that our estimator works, as the posterior distributions overlap well with the target values. This kind of model checking can be formalized as *simulation based calibration* [78], but even a few simulations are often sufficient to spot problems in conceptualising and programming the SCM, the statistical model, or both. The general idea is: until we are prepared to interpret estimates from synthetic data, we are unprepared to interpret estimates from real data.

The mapping between the structural and statistical parameters is, in the case of SCM 1, rather straightforward. Starting with $D$, we expect the statistical parameter to recover the intercept value of 0.2 (Eq 1.1.4). $G$, $R$, and $T$ are only caused by, respectively, $\gamma$, $\rho$, and $\tau$ (Fig 2B and Fig 3). Furthermore, $s_G$, $s_R$, and $s_T$, should capture the variation in $\gamma$, $\rho$, and $\tau$, respectively, and thus, be equal to 0.5 (see Eqs 1.1.1–1.1.3).

The correlation coefficient between $G$ and $R$, $c_{GR}$, does not have a counterpart in the SCM. However, looking at the DAG in Fig 2B, we can see that there is no path connecting $G_{[a]}$ and

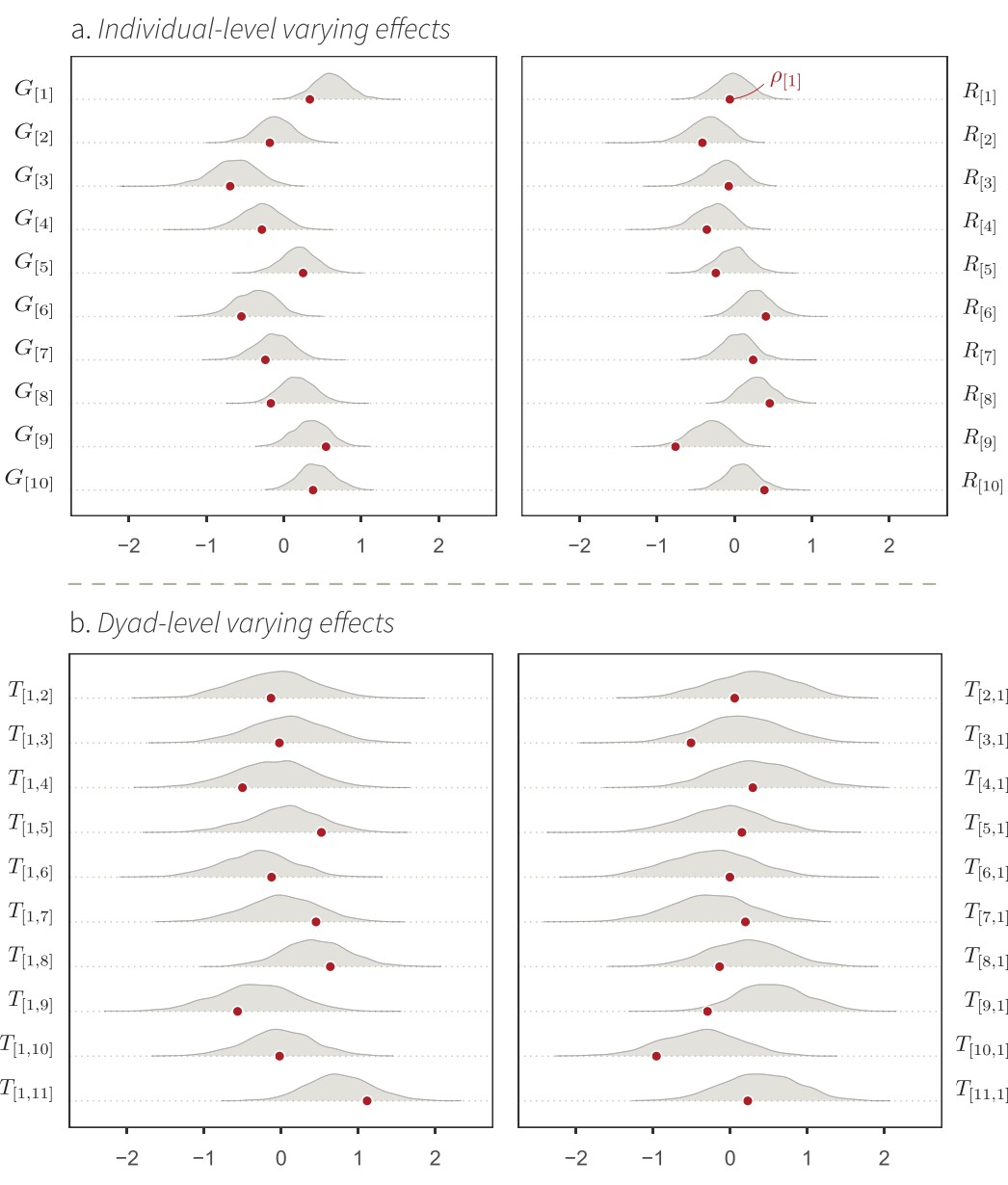

**Fig 3. Marginal posterior distributions of a set of statistical model 1's varying-effects.** The probability density of these parameters (*y-axis*) is shown for a range of parameter values (*x-axis*), for one iteration of the simulation. The red dots indicate the parameters' target values. For instance, $\gamma_{[1]}$ is the target value of $G_{[1]}$, whose posterior probability distribution is represented as a density curve on top of the red dot.

$R_{[a]}$ (or $G_{[b]}$ and $R_{[b]}$). Hence, there should be no association between them ($c_{GR} = 0$). The same reasoning applies to $c_{TT}$: there is no path between $T_{[a,b]}$ and $T_{[b,a]}$, and thus, we expect $c_{TT} = 0$. The Multivariate Normal posterior distribution describing $G_{[a]}$ and $R_{[a]}$, as well as the posterior distributions for $m$, are shown in Figs C–D of S1 Text.

**Conclusions.** In simulation 1, we have modelled a simple data-generating process, which describes the factors shaping the observed edges of a social network. We assumed that the general tendencies of individuals to interact with conspecifics, as well as their propensity to

engage in social interactions with specific partners, were random. We started by describing the abstract causal structure of this system, using a DAG. Next, we implemented this DAG as a SCM: a type of data simulation, where we specified parametric features of the causal system. We then fed the synthetic data into a statistical model, and showed that the statistical model could recover the structural parameters of the SCM. In the following section, we will turn to the following question: how can we use the models introduced in this section, to estimate the causal effect of individual-level features on their general propensity to engage in social interactions?

## Simulation study 2: Individual-level features

Here, we build upon the tools presented in the previous section to add features of individuals that may be targets of causal inference.

**Causal model.**  Suppose that we wish to study the effect of an individual-level phenotypic variable $X_{[a]}$ (e.g., age) on the overall tendency of individuals to give and receive social interactions (e.g., grooming), in a specific social system (e.g., a group of monkeys). We might, for instance, hypothesise that increasing age causes individual monkeys to generally disengage from their social group, and thus, to give and receive fewer grooming interactions [79,80]. Such a system may be represented as a DAG where $X_{[a]}$ has a direct effect on $\gamma_{[a]}$ and $\rho_{[a]}$ (see Fig 4A).

We wish to build an estimator that can estimate this effect. To do so, we specify quantitative causal assumptions using SCM 2. This model is identical to SCM 1, except with regard to the variable $X_{[a]}$ and its effects:

$$f_X: \quad X_{[a]} \sim \text{Normal}(0, 1). \tag{2.1.1}$$

$X_{[a]}$, an individual-level feature, is here defined as a standardised Gaussian distribution, with mean 0 and SD 1. Then,

$$f_\gamma: \quad \gamma_{[a]} \sim \text{Normal}(-0.7^* \cdot X_{[a]}, 0.5) \tag{2.1.2}$$

$$f_\rho: \quad \rho_{[a]} \sim \text{Normal}(-0.7^\ddagger \cdot X_{[a]}, 0.5). \tag{2.1.3}$$

These two equations encode the key aspects of simulation study 2: the causal effects of $X_{[a]}$ on $\gamma_{[a]}$ and $\rho_{[a]}$—and thus, on the rate $m_{[a,b]}$. These effects are, respectively, our two *estimands*; we mark them with ⋆ and ‡, such that they can be tracked more easily. $f_\gamma$ and $f_\rho$ imply that $\gamma_{[a]}$ and $\rho_{[a]}$ are now sampled from distributions, whose means depend on the value of $X_{[a]}$. To illustrate how this effect works, imagine two individuals, *a* and *b*, with $X_{[a]} = 1$ (old individual) and $X_{[b]} = -2$ (young individual). Consequently, $\gamma_{[a]}$ will be drawn from a distribution with mean $-0.7 \cdot 1 = -0.7$, and $\gamma_{[b]}$, from a distribution with mean $-0.7 \cdot (-2) = 1.4$. Thus, $\gamma_{[a]}$ will most likely be lower than $\gamma_{[b]}$, which is consistent with our hypothesis. Note that the effect of $X_{[a]}$ on $\gamma_{[a]}$ need not be the same as its effect on $\rho_{[a]}$.

Finally, $f_\tau$, $f_m$ and $f_y$, are the same as in SCM 1:

$$f_\tau: \quad \tau_{[a,b]} \sim \text{Normal}(0, 0.5) \tag{2.1.4}$$

$$f_m: \quad m_{[a,b]} = \exp(0.2 + \gamma_{[a]} + \rho_{[b]} + \tau_{[a,b]}) \tag{2.1.5}$$

$$f_y: \quad y_{[a,b]} \sim \text{Poisson}(m_{[a,b]}). \tag{2.1.6}$$

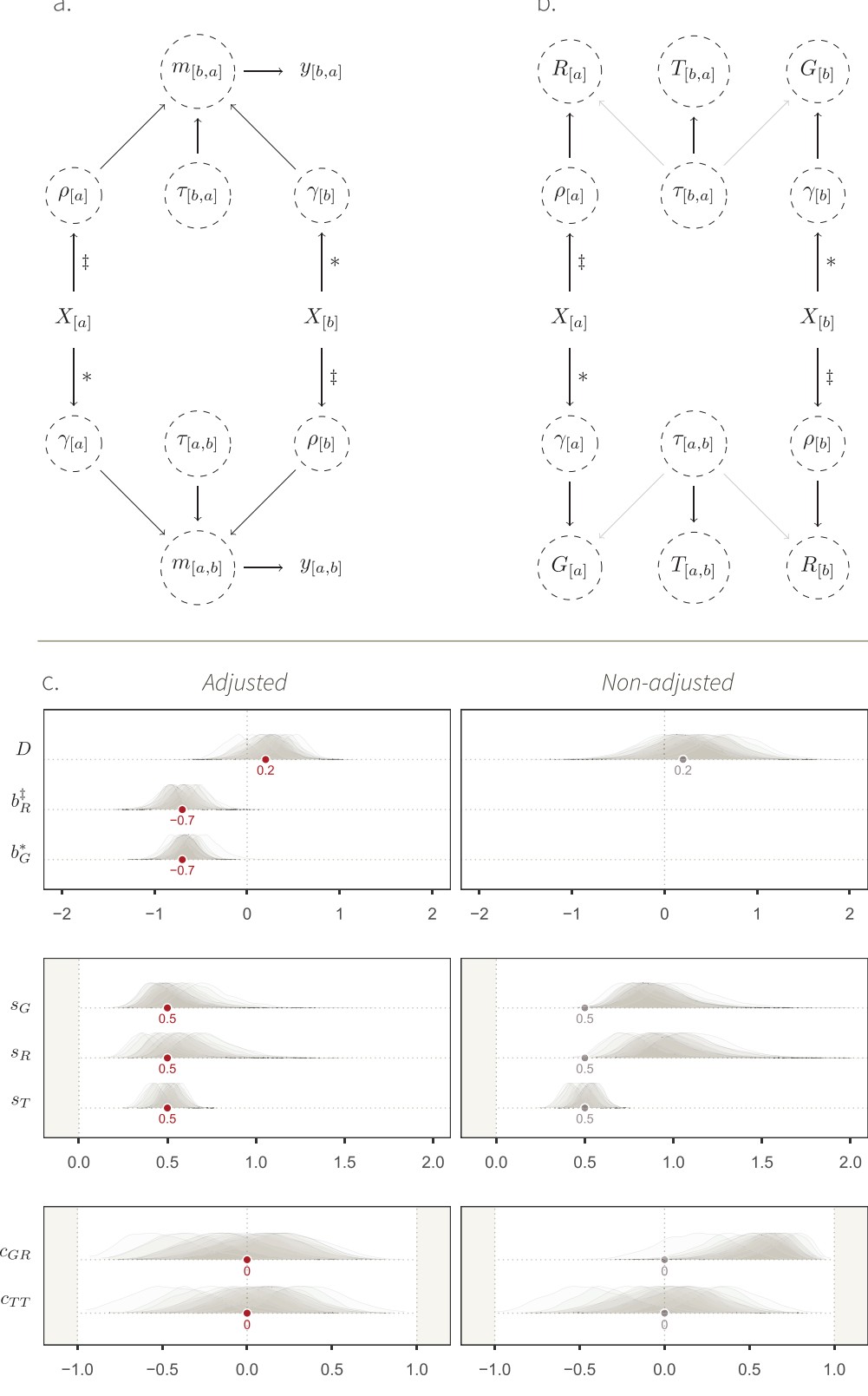

**Fig 4. Simulation study 2. A.** DAG describing a causal system where an individual-level phenotypic trait $X_{[a]}$, like age, affects their overall tendency to give interactions, $\gamma_{[a]}$, and their general tendency to receive interactions, $\rho_{[a]}$. The * and ‡ symbols mark our estimands: the causal effects we wish to estimate. The rest of the DAG is similar to Fig 2A. **B.** Mapping between structural parameters (SCM 2) and statistical parameters. As for Fig 2B, the transparent

arrows indicate that these effects are possible (e.g., simulation 3), but do not exist in SCM 2. **C.** Fixed-effect estimates, for two statistical models fitted to the data generated with SCM 2. *Left*: fixed effects of the social relations model adjusted by $X_{[a]}$ (statistical model 2). *Right*: fixed effects of the non-adjusted social relations model (statistical model 1). The target values of the well-adjusted model are shown in grey. Deviations from them allow us to understand how $X_{[a]}$ affects the varying effects of the model, when $X_{[a]}$ is not adjusted for.

The resulting network of (synthetic) observations $y_{[a,b]}$ is shown in Fig E of S1 Text.

**Statistical model.** Next, we describe *Statistical model 2*, an estimator for the causal effects of $X$ on $\gamma$ and $\rho$. This estimator maintains the basic architecture statistical model 1, but is slightly modified. Here, we define $\hat{G}_{[a]}$ and $\hat{R}_{[b]}$ as submodels of $m_{[a,b]}$, where $\hat{G}_{[a]}$ is stratified by the variables causing $\gamma_{[a]}$, and $\hat{R}_{[b]}$ is stratified by the variables causing $\rho_{[b]}$. Thus, both are stratified by $X$:

$$y_{[a,b]} \sim \text{Poisson}(m_{[a,b]}) \tag{2.2.1}$$

$$m_{[a,b]} = \exp(D + \hat{G}_{[a]} + \hat{R}_{[b]} + T_{[a,b]}) \tag{2.2.2}$$

$$\hat{G}_{[a]} = G_{[a]} + b_G^* \cdot X_{[a]} \tag{2.2.3}$$

$$\hat{R}_{[b]} = R_{[b]} + b_R^{\ddagger} \cdot X_{[b]}, \tag{2.2.4}$$

Where Eqs 2.2.2–2.2.4 are, together, mathematically equivalent to:

$$m_{[a,b]} = \exp(D + G_{[a]} + b_G^* \cdot X_{[a]} + R_{[b]} + b_R^{\ddagger} \cdot X_{[b]} + T_{[a,b]}).$$

We usually use the former notation, which is easier to read, and whose link with the non-adjusted Social Relations Model is more visible. Yet, these two sets of equations imply the same thing: $b_G^*$ is a slope estimating the association between $X_{[a]}$ and $m_{[a,b]}$—whether directly, or through a sub-model $\hat{G}_{[a]}$—, conditional on the other parameters of the model. Similarly, $b_R^{\ddagger}$ estimates the conditional association between $X_{[b]}$ and $m_{[a,b]}$. On the DAG, $b_G^*$ captures the path coefficient between $X_{[a]}$ and $m_{[a,b]}$; and $b_R^{\ddagger}$ the path between $X_{[b]}$ and $m_{[a,b]}$. Therefore, $b_G^*$ and $b_R^{\ddagger}$ should respectively recover the effect of $X_{[a]}$ on $\gamma_{[a]}$ (marked by *), and that of $X_{[b]}$ on $\rho_{[b]}$ (marked by a ‡), and take a value of –0.7.

Finally, $G_{[a]}$, $R_{[b]}$, and $T_{[a,b]}$ are varying effects. They are part of the exact same multivariate adaptive priors described in statistical model 1:

$$\begin{pmatrix} G_{[a]} \\ R_{[a]} \end{pmatrix} \sim \text{MVNormal}\left[ \begin{pmatrix} 0 \\ 0 \end{pmatrix}, \begin{pmatrix} s_G^2 & c_{GR}s_Gs_R \\ c_{GR}s_Gs_R & s_R^2 \end{pmatrix} \right] \tag{2.2.5}$$

$$\begin{pmatrix} T_{[a,b]} \\ T_{[b,a]} \end{pmatrix} \sim \text{MVNormal}\left[ \begin{pmatrix} 0 \\ 0 \end{pmatrix}, \begin{pmatrix} s_T^2 & c_{TT}s_T^2 \\ c_{TT}s_T^2 & s_T^2 \end{pmatrix} \right]. \tag{2.2.6}$$

The full model with hyperpriors can be found in Sect B.2 of S1 Text. We also wish to describe how $X_{[a]}$ would impact the varying effects of the statistical model if $X_{[a]}$ was not observed. Thus, we also fitted the data generated with SCM 2 into statistical model 1.

**Posterior model.** Statistical model 2 successfully recovered the structural parameters of the simulation (Fig 4C, left panel). Most importantly, we accurately estimated our estimands: $b_G^*$ recovered the true causal effects of $X_{[a]}$ on $\gamma_{[a]}$, and $b_R^{\ddagger}$, the effect of $X_{[a]}$ on $\rho_{[a]}$.

This means that our estimator is capable of producing valid causal inference under the assumptions embodied by the DAG and the SCM.

On the right panel of Fig 4C, we observe important deviations between, on the one hand, the target values of the adjusted estimator, and, on the other hand, the posterior distributions of the non-adjusted statistical model. The posterior distributions of $s_G$ and $s_R$—respectively quantifying the variation in $G_{[a]}$ and $R_{[a]}$—are now consistently higher than 0.5. This is because $G_{[a]}$ and $R_{[a]}$ are caused by $\gamma_{[a]}$ and $\rho_{[a]}$, which are themselves caused by *two* kinds of variables. They are affected by random noise, quantified by a Gaussian SD of 0.5, and by $X_{[a]}$ (see Eq 2.1.2–2.1.3). Hence, $s_G$ and $s_R$ are larger than 0.5 if we don't stratify by $X_{[a]}$ (right panel), but are equal to 0.5 once we do condition on—or "control for"—it (left panel).

Furthermore, the posterior distribution of $c_{GR}$ is positive in the non-adjusted model (Fig 4; see Fig F in S1 Text for an additional visualisation). Recall that $c_{GR}$ captures the association between $G_{[a]}$ and $R_{[a]}$. Looking at the DAG, we see that these parameters are connected by $X_{[a]}$. $X_{[a]}$ creates an open path,

$$G \leftarrow \gamma \leftarrow X \rightarrow \rho \rightarrow R,$$

Which lets the association flow between $G_{[a]}$ and $R_{[a]}$. Once we condition on $X_{[a]}$, however, we block this path, cancel the association between $G_{[a]}$ and $R_{[a]}$, and we return to a correlation coefficient $c_{GR} = 0$ (left panel). Finally, notice that the dyadic parameters $s_T$ and $c_{TT}$, were unaffected by $X_{[a]}$. This can be deduced from the DAG, too, for there exist no open paths connecting $\gamma_{[a]}$ or $\rho_{[a]}$ — carrying the effects of $X_{[a]}$—and $T_{[a,b]}$.

**Conclusions.** In simulation 2, we have modelled a data-generating process where an individual-level trait $X_{[a]}$ (e.g., age) affected the general tendency of individuals to give and receive social interactions. Following the same workflow as in simulation 1, we showed that a well-specified estimator—an adjusted version of the Social Relations Model—could accurately recover the causal effect of $X_{[a]}$ on social network structure. Next, we highlighted how the causal effect of $X_{[a]}$ impacted the varying-effects structure of the statistical model, when $X_{[a]}$ was not adjusted for. We refer the interested readers to the supplementary Sect C in S1 Text: there, we describe *simulation study 2'*, a variation of simulation study 2, where $X_{[a]}$ is a categorical variable (e.g., sex). In the next section, we turn to the following question: how can we build a statistical model to estimate the effect of dyad-level features on the propensity of two individuals to socially interact with one another?

## Simulation study 3: Dyad-level features

Suppose that we wish to study how the genetic relatedness of individuals affects the way they interact with one another. We may hypothesise that individuals belonging to the same kin group engage in a greater number of affiliative interactions than non-kin. Furthermore, sampling effort may vary across individuals and dyads, for behavioural observation is conducted by following focal animals using standardised protocols. Individuals of some dyads might have been observed for several hours of behavioural sampling, while others might have been followed for very little time.

**Causal model.** We represent the causal structure of such a data-generating process on a DAG (Fig 5A), where we specify these causal relationships, as well as the link between individual- and dyad-level features. The combinations of two individuals' kin groups, $K_{[a]}$ and $K_{[b]}$, determines how genetically related they are to one another ($Re_{|a,b|}$), by definition. For the dyad $[a,b]$ to be observed, one has to observe either $a$ or $b$; thus, the sampling effort for

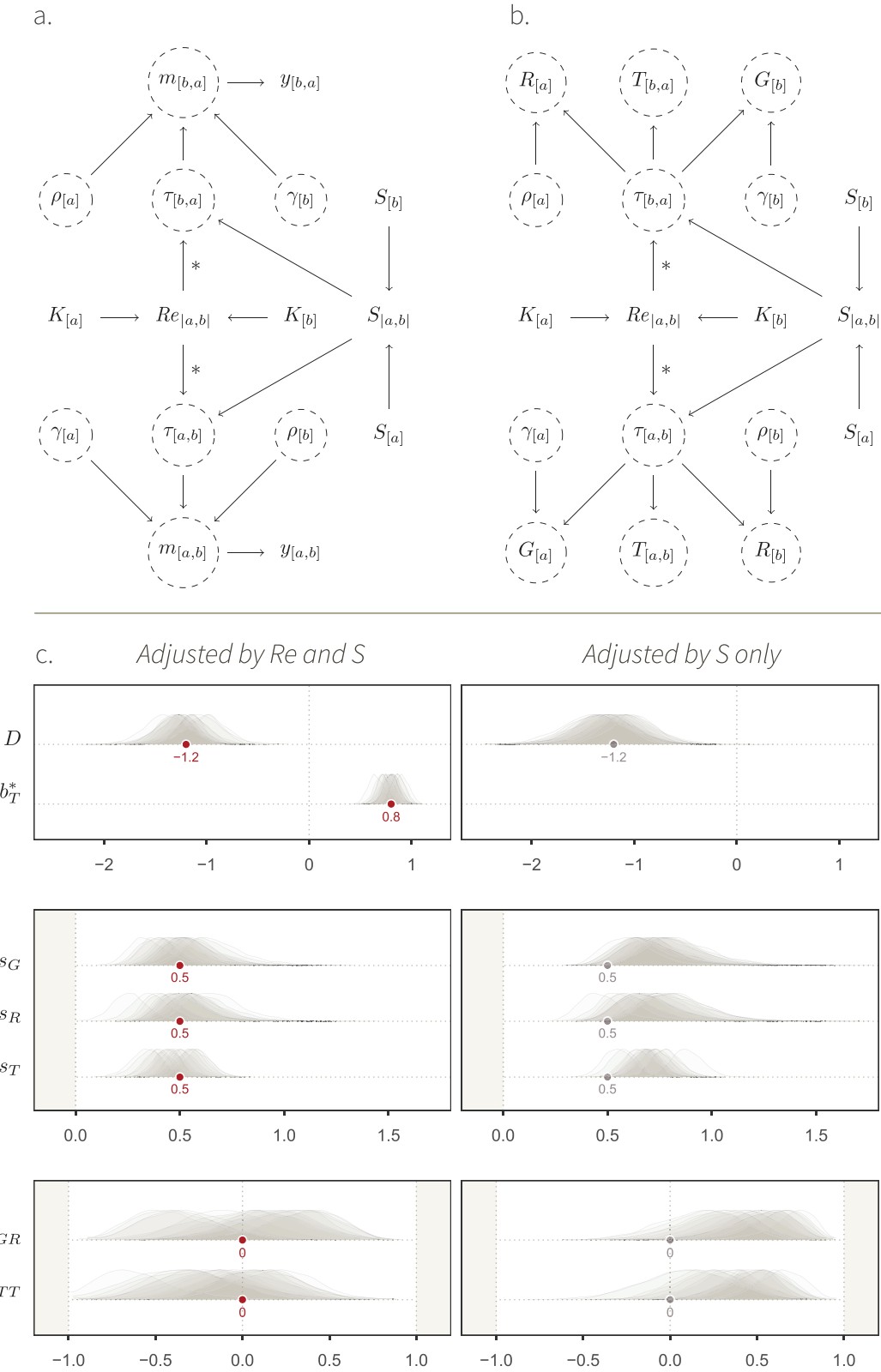

**Fig 5. Simulation study 3. A.** DAG representing a causal system where the kin group $K_{[a]}$ of individuals affect how they preferentially interact with one another. Individuals belonging to the same kin group have a higher relatedness $Re_{|a,b|}$ than non-kin. $Re_{|a,b|}$, in turn, affects how the pair of individuals interact with one another, through $\tau_{[a,b]}$. This effect is our estimand, and we mark it with *. The sampling effort of the two individuals, $S_{[a]}$ and $S_{[b]}$, determine $S_{|a,b|}$, which in

turn affects the unit of $m_{[a,b]}$. The rest of the DAG is similar to Fig 2A. **B.** Mapping between structural parameters (SCM 3) statistical parameters. Compared to the simulation studies 1 and 2 where the only cause of $\tau_{[a,b]}$ was exogenous noise, $\tau_{[a,b]}$ is here affected by $Re_{|a,b|}$ and $S_{|a,b|}$, and it has an effect on $G_{[a]}$ and $R_{[b]}$. **C.** *Left*: fixed effects of the social relations model adjusted by $Re_{|a,b|}$ and $S_{|a,b|}$ (statistical model 3). *Right*: fixed effects of the social relations model adjusted by $S_{|a,b|}$ only. The target values of the well-adjusted model are shown in grey. Deviations from them allow us to understand how $Re_{|a,b|}$ affects varying effects, when $Re_{|a,b|}$ is not adjusted for.

each individual, $S_{[a]}$, in combination with the sampling effort for its partner, $S_{[b]}$, deterministically cause the dyad-level sampling effort, $S_{|a,b|}$—quantifying how long either $a$ or $b$ was observed for—, which in turns affects $m_{[a,b]}$ through $\tau_{[a,b]}$ (for a discussion on deterministic arrows, see [81,82]). Note that there is no path between $Re_{|a,b|}$ and $m_{[a,b]}$ that passes through $S_{|a,b|}$. This means that $S_{|a,b|}$ does not *confound* the effect of $Re_{|a,b|}$. Yet, we still want to include it in our estimator (see next section), for doing so will increase the model's precision.

Before building the estimator, we translate the assumptions of the DAG into *SCM 3*. We simulate 20 individual animals, belonging to 11 different kin groups: 10 individuals belong to one, large, kin group, and each of the 10 remaining individuals are alone in their kin group (Fig I in S1 Text). For simplicity, we dichotomise genetic relatedness, such that:

$$f_{Re}: \quad Re_{|a,b|} = \begin{cases} 1 & \text{if } K_{[a]} = K_{[b]} \\ 0 & \text{if } K_{[a]} \neq K_{[b]} \end{cases} \tag{3.1.1}$$

We further assume that the variation in sampling effort is random:

$$f_{S_{[a]}}: \quad S_{[a]} \sim \text{Uniform}(1, 2.5). \tag{3.1.2}$$

The observation effort per individual takes a value between 1 and 2.5 time units—e.g., the number of full days of focal-follows. We then combine the sampling effort of the two interacting individuals into their dyad-level sampling effort:

$$f_{S_{|a,b|}}: \quad S_{|a,b|} = S_{[a]} + S_{[b]}. \tag{3.1.3}$$

Then, we define how different variables affect $\gamma$, $\rho$, and $\tau$:

$$f_{\gamma}: \quad \gamma_{[a]} \sim \text{Normal}(0, 0.5) \tag{3.1.4}$$

$$f_{\rho}: \quad \rho_{[a]} \sim \text{Normal}(0, 0.5) \tag{3.1.5}$$

$$f_{\tau}: \quad \tau_{[a,b]} \sim \text{Normal}(\log(S_{|a,b|}) + 0.8^* \cdot \tilde{Re}_{|a,b|}, 0.5), \tag{3.1.6}$$

Where the $\sim$ symbol on top of $\tilde{Re}_{|a,b|}$ means that we standardised it using a z-transformation.

$$f_{m}: \quad m_{[a,b]} = \exp(-1.2 + \gamma_{[a]} + \rho_{[b]} + \tau_{[a,b]}) \tag{3.1.7}$$

$$f_{y}: \quad y_{[a,b]} \sim \text{Poisson}(m_{[a,b]}). \tag{3.1.8}$$

Compared to the SCMs above, the intercept is set to –1.2, to make the interaction rates similar to those in the simulation studies above. In SCM 3, an unrelated average dyad observed for 3.5 time units, would have an expected value of $m_{[a,b]} = \exp(-1.2 + \log(3.5)) \simeq$ 1.1 interactions, compared to $m_{[a,b]} = \exp(0.2) \simeq 1.2$ interactions in the previous simulation studies. We show the resulting network of synthetic observations, where the effect of genetic relatedness on social structure is visible by eye, in Fig I in S1 Text.

To illustrate how the offset $\log(S_{|a,b|})$ scales the rates $m_{[a,b]}$—through its effect on $\tau_{[a,b]}$—, imagine two average, directed dyads [1,2] and [12,9], each composed of non-kin. Assume their respective rate of interaction per *one* time unit is identical, and is equal to: $\exp(-1.2 + \log(1)) = \exp(-1.2) = 0.3$. Let $S_{|1,2|} = 4$ and $S_{|12,9|} = 2$. Consequently, the dyads' interaction rates will be scaled by their respective values of $S_{|a,b|}$, such that: $m_{[1,2]} = \exp(-1.2 + \log(4)) =$ 1.2 and $m_{[12,9]} = \exp(-1.2 + \log(2)) = 0.6$. Their meaning, thus, slightly differs from one another: $m_{[1,2]}$ is now the average number of interactions per *four* times units, and $m_{[12,9]}$ is the average number of interactions per *two* time units. In Sect E of S1 Text, we show an alternative parameterisation of SCM 3 model where $S_{|a,b|}$ affects $y_{[a,b]}$ directly, instead of scaling $m_{[a,b]}$ through $\tau_{[a,b]}$.

**Statistical model.**   Below, we describe *Statistical Model* 3, an estimator for the effect of genetic relatedness $Re_{|a,b|}$ on $m_{[a,b]}$.

$$y_{[a,b]} \sim \text{Poisson}(m_{[a,b]}) \tag{3.2.1}$$

$$m_{[a,b]} = \exp(D + G_{[a]} + R_{[b]} + \hat{T}_{[a,b]}). \tag{3.2.2}$$

Like in statistical model 1, $G_{[a]}$ and $R_{[a]}$ are part of a Multivariate adaptive prior distribution,

$$\begin{pmatrix} G_{[a]} \\ R_{[a]} \end{pmatrix} \sim \text{MVNormal}\left[ \begin{pmatrix} 0 \\ 0 \end{pmatrix}, \begin{pmatrix} s_G^2 & c_{GR}s_G s_R \\ c_{GR}s_G s_R & s_R^2 \end{pmatrix} \right]. \tag{3.2.3}$$

$\hat{T}_{[a,b]}$ is, however, described by a submodel, stratified by the causes of $\tau_{[a,b]}$:

$$\hat{T}_{[a,b]} = \log(S_{|a,b|}) + T_{[a,b]} + b_T^* \cdot \tilde{R}e_{|a,b|} \tag{3.2.4}$$

$$\begin{pmatrix} T_{[a,b]} \\ T_{[b,a]} \end{pmatrix} \sim \text{MVNormal}\left[ \begin{pmatrix} 0 \\ 0 \end{pmatrix}, \begin{pmatrix} s_T^2 & c_{TT}s_T^2 \\ c_{TT}s_T^2 & s_T^2 \end{pmatrix} \right]. \tag{3.2.5}$$

The submodel for $\hat{T}_{[a,b]}$ is offsetted by $\log(S_{|a,b|})$, which works like the offset of SCM 3. By making dyads comparable to one another, this term naturally deals with the issue of sampling biases that we introduced earlier (problem II). $b_T^*$, accordingly, captures the association between genetic relatedness and $m_{[a,b]}$, *conditional on* the variation in $S_{|a,b|}$ among dyads. This parameter should recover our estimand by taking a value of 0.8. The full specification of statistical model 3 can be found in Sect D.2 of S1 Text. In addition to statistical model 3, we also fitted the data generated by SCM 3 to a version of statistical model 3 that was not adjusted by $Re_{|a,b|}$. That is, Eq 3.2.4 was replaced by:

$$\hat{T}_{[a,b]} = \log(S_{|a,b|}) + T_{[a,b]}.$$

**Posterior model.** Statistical model 3 successfully recovered the structural parameters of the generative model, SCM 3 (Fig 5C, left panel). Most notably, $b_T^*$, the statistical parameter whose aim was to estimate our causal estimand, accurately recovered the value of 0.8. This result validates our estimator, given the causal assumptions encoded in the DAG and SCM.

Below, we briefly explain how the effect of genetic relatedness, a *dyadic* variable, results in variation in *individual-* and *dyad*-level varying effects, when relatedness is not included in the statistical model (Fig 5C, right panel). In simulation study 1 and 2, the structural and statistical parameters were equivalent: $\gamma_{[a]}$ was the only cause of $G_{[a]}$, $\rho_{[a]}$ the only cause of $R_{[a]}$, and $\tau_{[a,b]}$ the only cause of $T_{[a,b]}$ (e.g., Fig 4). As a result, $G_{[a]}$ fully recovered $\gamma_{[a]}$, $T_{[a,b]}$ recovered $\tau_{[a,b]}$, etc., such that the parameters capturing their patterns of (co)variation—$s_G$, $s_R$, $c_{GR}$—could be directly mapped onto the structural model. However, this is not the case here. We see that the estimates for $s_G$, $s_R$ and $c_{GR}$ substantially deviate from the grey dots—i.e., the value they wound take if none of the effect of $\tau_{[a,b]}$ "leaked" onto $G_{[a]}$ and $R_{[b]}$.

Recall that $G_{[a]}$, $R_{[b]}$, and $T_{[a,b]}$ are *statistical* parameters, meaning they only measure (conditional) averages. They cannot "see" that the effect was encoded as a dyadic effect through $\tau_{[a,b]}$. Instead, $G_{[a]}$ and $R_{[b]}$ measure interindividual differences in average interaction rates. Genetic relatedness, although it acts at the dyadic level, creates such differences. Individuals belonging to the large kin group give, and receive, more frequent social interactions, than those in the small kin groups (see Fig 5C, right panel, and Fig I in S1 Text). This interindividual variation is visible as an increase in $s_G$ and $s_R$, and in covariation between these two individual features, indicated by a positive value for $c_{GR}$. This pattern can also be explained with the DAG (Fig 5B): there is an open path between $G_{[a]}$ and $R_{[a]}$, passing through $Re_{|a,b|}$. The effect of $Re_{|a,b|}$ that can be attributed to interindividual variation is absorbed by $G_{[a]}$ and $R_{[a]}$, and the *residual* variation is absorbed by $T_{[a,b]}$. This residual variation does *not* correspond to a process in the SCM, and therefore, $s_T$ and $c_{TT}$ do not have a straightforward biological interpretation. Once we condition on $Re_{|a,b|}$, we block the paths between $G_{[a]}$ and $R_{[a]}$, and between $T_{[a,b]}$ and $T_{[b,a]}$. Hence, the correlations captured by $c_{GR}$ and $c_{TT}$ disappear (Fig 5C, left panel).

**Conclusions.** In simulation study 3, we modelled a data-generating process where two dyad-level variables affected the observed social interactions among individuals. Genetic relatedness, a biological variable, affected the rate at which individuals interacted with one another; it was our estimand. Sampling effort, a variable describing the sampling process, further impacted the scale of the interaction rates. We then built our estimator in combination with a DAG and a SCM, and successfully recovered the true causal effect. Next, we showed that dyad-level causal effects could impact individual- and dyad-level varying parameters, when not accounted for. Once again, we refer the interested readers to simulation study 3', a variation of simulation study 3, where we estimate the causal effect of a categorical dyad-level variable, like the combination of sexes, using a stochastic block model structure (Sect F in S1 Text). In the next section, building upon our previous models, we ask: how can we develop a statistical model to estimate the effect of a dyad-level variable if this effect is *confounded* (problem III)?

## Empirical example

In this section, we showcase the use of our framework to address a causal question in a specific empirical system. Our aim is to estimate the effect of maternal relatedness on affiliative behaviour in the females of a wild population of Assamese macaques (*Macaca assamensis*). The causal assumptions that we will make about this social system are, although crude, reasonably realistic. Thus we will conclude the section by fitting our estimator to an empirical

dataset, and we will show how to compute causal estimates on the outcome scale from the joint posterior probability distribution.

## Simulation study 4: Kinship in female macaques

**Verbal description.** Kinship in female macaques, like in other Cercopithecinae species, is thought to be a major driver of social network structure [14,83,84]. The kin group of a female macaque not only determines who she is genetically related to, it also affects her position in the group's dominance hierarchy. Females (non-genetically) inherit their rank from their mothers, such that individuals in the same kin group generally occupy adjacent ranks. Furthermore, kinship affects the formation of the social groups themselves. When groups of macaques permanently split, they usually divide along matrilines, and members of the same matrilines stay together in the newly formed groups [85–87]. This phenomenon may result in smaller social groups—often, newly formed ones—to contain fewer kin groups, and thus to have a higher average degree of genetic relatedness.

The genetic relatedness between two individuals, as well as their respective positions in the dominance hierarchy, might both affect the pattern of affiliative interactions they exchange with one another [8]. As suggested earlier, genetically related individuals might exchange more affiliative interactions because of a preference for their kin. Additionally, dominant females may be attractive affiliation partners. Individuals might, accordingly, preferentially target their dominants—rather than their subordinates—with affiliative interactions.

**Directed acyclic graph.** We represent this causal system as a DAG (Fig 6A), and assume that the rate of grooming bouts $m_{[a,b]}$ is a good proxy for affiliative relationships [15]. The causal graph shares its basic structure with that of simulation study 3. We add an effect of kin group $K_{[a]}$ on individual rank $Ra_{[a]}$. $Ra_{[a]}$, in combination with $Ra_{[b]}$ then determine the *rank difference* between two individuals, $\Delta Ra_{|a,b|}$, which in turn affects $\tau_{[a,b]}$. Finally, we represent group size $GS_{\text{gr}_{|a,b|}}$ as a group-level feature (hence the "gr" index). The double-headed arrow between $K$ and $GS_{\text{gr}}$ means that they are both connected by an open path. It simply encodes that larger social groups tend to contain more kin groups while remaining agnostic about the complex, underlying, causal mechanism (i.e. group fission).

An important insight provided by the DAG is that our estimand is confounded by dominance rank (Fig 6A). There are now two paths connecting $Re_{|a,b|}$ and $m_{[a,b]}$. There is, first, a "frontdoor" path, transmitting the causal effect of interest:

$$Re \rightarrow \tau \rightarrow m,$$

Second, there is a "backdoor" path transmitting a spurious association between $Re$ and $m$:

$$Re \leftarrow K \rightarrow Ra \rightarrow \tau \rightarrow m.$$

If we were to regress $m_{[a,b]}$ on $Re_{|a,b|}$ to estimate our estimand without stratifying by rank (as we did in simulation study 3), our effect would be biased by the backdoor path. Note that this would happen even with an infinitely large data set: the causal graph informs us that, because of the backdoor path, our effect simply *cannot* be estimated by looking at the raw association between $Re_{|a,b|}$ and $m_{[a,b]}$. To develop a better intuition for this issue, we first translate this DAG into SCM 4.

**Structural causal model.** In SCM 4, we create three groups of 10, 15, and 20 individuals, which vary in their maximum number of kin groups $N_{K_{\text{gr}_{|a,b|}}}$: 3, 5, 7, respectively. Each individual's kin group $K_{[a]}$ is drawn with replacement from the vector $\{1, \dots, N_{K_{\text{gr}_{[a]}}}\}$. Doing so

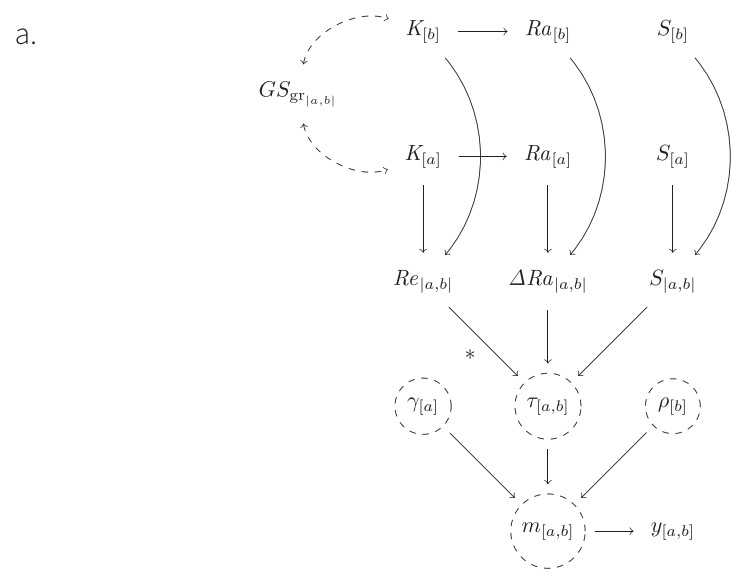

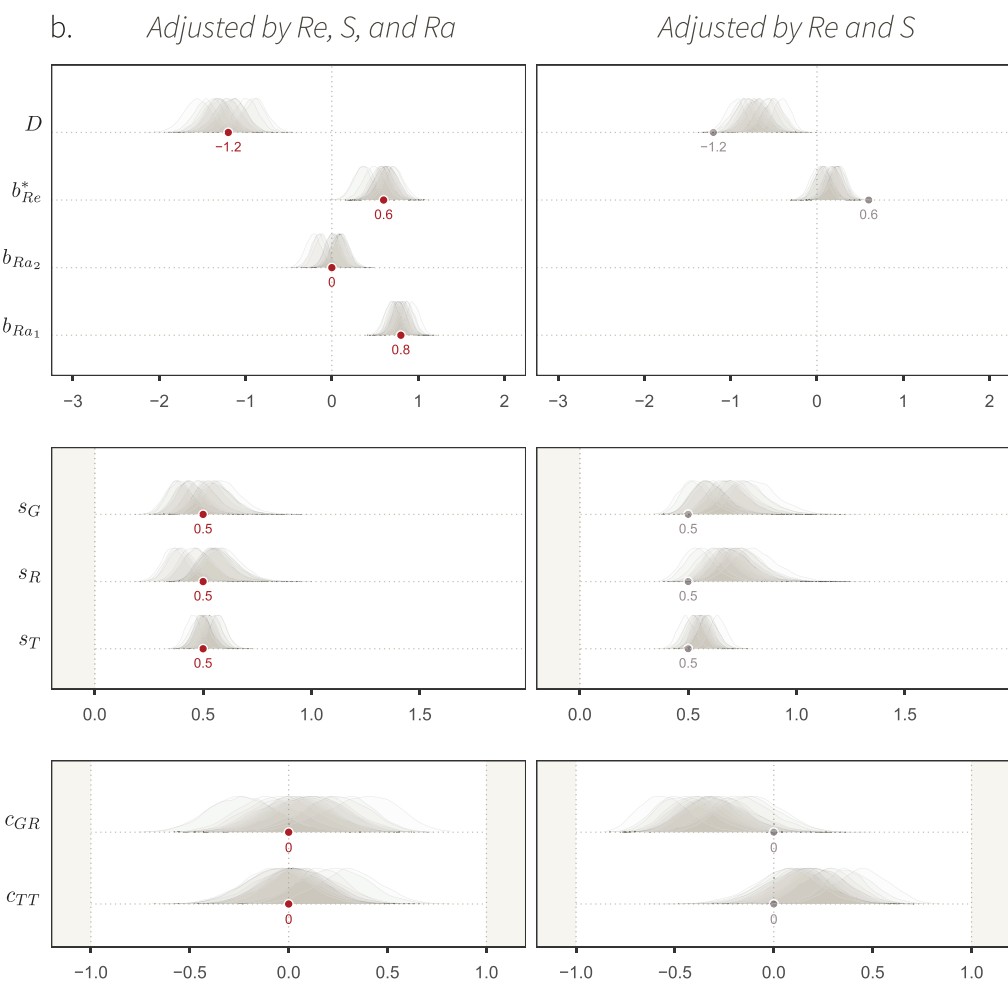

**Fig 6. Simulation study 4. A.** DAG for the dual effect of kinship on grooming interactions in macaques. As in simulation study 3, the kin groups of two individuals, $K_{[a]}$ and $K_{[b]}$, determine their degree of genetic relatedness $Re_{|a,b|}$, which in turn affects $\tau_{[a,b]}$. This effect is our estimand, which we mark with *. $K_{[a]}$ and $K_{[b]}$ affect the dominance ranks $Ra_{[a]}$ and $Ra_{[b]}$ of the individuals, whose difference $\Delta Ra_{|a,b|}$ affect $\tau_{[a,b]}$. Dyads further vary in their respective

sampling effort $S_{|a,b|}$. Finally, the size of social groups $GS_{\mathrm{gr}_{|a,b|}}$ covaries with the number of kin groups $K$ within in. The rest of the DAG is similar to Fig 5A. **B.** *Left*: fixed effects of the social relations model adjusted by $Re_{|a,b|}$, $S_{|a,b|}$ and $\Delta Ra_{|a,b|}$ (statistical model 4). *Right*: fixed effects of the social relations model adjusted by $Re_{|a,b|}$ and $S_{|a,b|}$ only. The target values of the well-adjusted model are shown in grey. Deviations from them allow us to understand how $\Delta Ra_{|a,b|}$ counfounds our estimand, when $\Delta Ra_{|a,b|}$ is not adjusted for.

leads to a higher degree of average genetic relatedness in smaller social groups than in larger ones (double arrow in Fig 6). The kin groups are then assigned kin-group dominance rank $Ra_{K_{[a]}}$ (i.e. a rank for the whole matriline):

$$f_{Ra_K}: \quad Ra_{K_{[a]}} = K_{[a]}. \tag{4.1.1}$$

That is, kin groups are assigned a dominance rank at random. A lower $Ra_{K_{[a]}}$ imply a higher rank. Thus, if $Ra_{K_{[a]}} < Ra_{K_{[b]}}$, then the dominance rank of all of $a$'s kin is higher than that of all of $b$'s kin. Finally, within each kin group, individuals receive ranks at random too. Overall, these steps represent an implementation of the arrow from $K$ to $Ra$, resulting in a dominance hierarchy that is stratified by matrilines of various sizes.

Next, sampling effort varies across individuals and dyads:

$$f_{S_{[a]}}: \quad S_{[a]} \sim \mathrm{Uniform}(1, 2.5) \tag{4.1.2}$$

$$f_{S_{|a,b|}}: \quad S_{|a,b|} = S_{[a]} + S_{[b]}. \tag{4.1.3}$$

Sampling effort, genetic relatedness, and dominance rank difference all affect the interaction rate through $\tau_{[a,b]}$:

$$f_{\tau}: \quad \tau_{[a,b]} \sim \begin{cases} \mathrm{Normal}(\log(S_{|a,b|}) + 0.6^* \cdot Re_{|a,b|} \\ \quad + 0.8 \cdot (Ra_{[b]} - Ra_{[a]}), 0.5) & \text{if } R_{[a]} < R_{[b]} \\ \mathrm{Normal}(\log(S_{|a,b|}) + 0.6^* \cdot Re_{|a,b|} \\ \quad + 0 \cdot (Ra_{[b]} - Ra_{[a]}), 0.5) & \text{if } R_{[a]} \geq R_{[b]} \end{cases} \tag{4.1.4}$$

The main novelty of $f_{\tau}$, compared to SCM 3, regards the causal effect of rank; or rather, the effect of the difference in ranks $Ra_{[b]} - Ra_{[a]}$. Here, individuals increase their interaction rate as a function of how much higher in the hierarchy their partner is: a larger increase for a larger difference. The effect is asymmetrical, and individuals are not impacted by how much lower in the hierarchy their partner is. The remaining structural equations are identical to those in SCM 3:

$$f_{\gamma}: \quad \gamma_{[a]} \sim \mathrm{Normal}(0, 0.5) \tag{4.1.5}$$

$$f_{\rho}: \quad \rho_{[a]} \sim \mathrm{Normal}(0, 0.5) \tag{4.1.6}$$

$$f_m: \quad m_{[a,b]} = \exp(-1.2 + \gamma_{[a]} + \rho_{[b]} + \tau_{[a,b]}) \tag{4.1.7}$$

$$f_y: \quad y_{[a,b]} \sim \mathrm{Poisson}(m_{[a,b]}). \tag{4.1.8}$$

The network of simulated observations $y_{[a,b]}$, as well as their distribution, are shown in Fig N in S1 Text.

Coming back to our backdoor problem, let us now imagine two directed dyads: [1,2] and [3,4]. We assume that the exogenous (random) causes of $m_{[1,2]}$ and $m_{[3,4]}$ are set to zero.

Suppose that the first dyad is composed of kin ($Re_{|1,2|} = 1$) who occupy adjacent ranks in the dominance hierarchy ($\Delta Ra_{|1,2|} \simeq 0$). Thus,

$$m_{[1,2]} = \exp(-1.2 + 0.6^* \cdot 1 + 0.8 \cdot 0) \simeq 0.5.$$

Suppose that the second dyad is composed of non-kin ($Re_{|3,4|} = 0$), who are further apart in the dominance hierarchy: $\Delta Ra_{|3,4|} \simeq 0.7$. Therefore,

$$m_{[3,4]} = \exp(-1.2 + 0.6^* \cdot 0 + 0.8 \cdot 0.7) \simeq 0.5.$$

Despite the positive effect of relatedness on grooming rate, $m_{[1,2]} \simeq m_{[3,4]}$, because the two dyads were not comparable to one another with respect to rank. The same applies in the rest of the population, where kin dyads are closer in the dominance hierarchy on average compared to non-kin dyads, thereby making the effect of relatedness unidentifiable by directly comparing kin and non-kin. Instead, to identify the effect of interest, we need to block the path created by rank by conditioning on it (we apply the backdoor criterion; see [67]). That is, to estimate our estimand, we need to build an estimator that can address the following question: "*once we know the difference in rank between a and b, what association remains between genetic relatedness and grooming rate?*".

**Statistical model.** Statistical model 4 is identical to statistical model 3, except that it includes dominance rank.

$$y_{[a,b]} \sim \text{Poisson}(m_{[a,b]}) \tag{4.2.1}$$

$$m_{[a,b]} = \exp(D + G_{[a]} + R_{[b]} + \hat{T}_{[a,b]}) \tag{4.2.2}$$

$$\hat{T}_{[a,b]} = \log(S_{|a,b|}) + T_{[a,b]} + b_{Re}^* \cdot Re_{|a,b|}$$

$$+ \begin{cases} b_{Ra_1} \cdot (Ra_{[b]} - Ra_{[a]}) & \text{if } R_{[a]} < R_{[b]} \\ b_{Ra_2} \cdot (Ra_{[b]} - Ra_{[a]}) & \text{if } R_{[a]} \geq R_{[b]} \end{cases} \tag{4.2.3}$$

The stratification by $\Delta Ra_{|a,b|}$ allows for asymmetrical effects. $b_{Ra_1}$ should recover the effect of 0.8, and $b_{Ra_2}$ should recover 0 (Eq 4.1.4). Conditioning on rank closes the backdoor path mentioned earlier, and thus, $b_{Re}^*$ should recover the unbiased effect of 0.6. Finally, $G_{[a]}$, $R_{[a]}$, and $T_{[a,b]}$ are varying-effects, as in SCM 3:

$$\begin{pmatrix} G_{[a]} \\ R_{[a]} \end{pmatrix} \sim \text{MVNormal}\left[ \begin{pmatrix} 0 \\ 0 \end{pmatrix}, \begin{pmatrix} s_G^2 & c_{GR}s_G s_R \\ c_{GR}s_G s_R & s_R^2 \end{pmatrix} \right] \tag{4.2.4}$$

$$\begin{pmatrix} T_{[a,b]} \\ T_{[b,a]} \end{pmatrix} \sim \text{MVNormal}\left[ \begin{pmatrix} 0 \\ 0 \end{pmatrix}, \begin{pmatrix} s_T^2 & c_{TT}s_T^2 \\ c_{TT}s_T^2 & s_T^2 \end{pmatrix} \right]. \tag{4.2.5}$$

The full specification of the model, as well as prior predictive simulations, can be found in Sects G.2 and G.3 of S1 Text. We also fitted the synthetic data to an estimator that was *not* stratified by rank—i.e. without the second and third lines of Eq 4.2.3—, so that we could visualise the confounding effect of the backdoor path.

**Posterior model (simulated data).** Statistical model 4 successfully recovered our estimand by deconfounding the biasing effect of dominance rank (Fig 6B). Looking at the left panel, we see that the marginal posterior distributions of $b_{Re}^*$ are concentrated on the target value of 0.6. This is not the case for the panel on the right, where the model that was not adjusted by rank gives a consistently biased estimate, close to 0. These results confirm that

given our causal assumptions, a statistical model that correctly blocks the backdoor path between $Re_{|a,b|}$ and $m_{[a,b]}$ is *necessary* to accurately estimate our estimand. More generally, the results validate our estimator for the effect of genetic relatedness in female Assamese macaques; insofar, at least, as we believe that our causal assumptions are good approximations of the data-generating process (we will come back to this issue in the discussion). We can thus turn to updating this estimator with empirical data.

## Empirical data

The empirical observations that we fit to statistical model 4 were collected as part of a long-term research project on wild Assamese macaques at Phu Khieo Wildlife Sanctuary, in Northeastern Thailand (Fig 7A). We focused on the adult females of three social groups. The data collection took place between July 2017 and July 2018. The animals were fully habituated to the presence of researchers. Observers recorded the monkeys' behaviour using continuous focal sampling protocols of 40 minutes, during which they recorded all instances of dyadic interactions between adult females, including grooming and submissive behaviours, from which dominance ranks were computed. Submissive behaviours were also recorded *ad libitum*.

We defined $S_{[a]}$ as the number of 12 hours periods that each individual was observed for (hereafter "days" of observations), summed over sampling protocols. The average sampling effort was equal to $6 \pm 1.4$ days per individual (Fig R in S1 Text). As in SCM 4, $S_{|a,b|} = S_{[a]} + S_{[b]}$ (Fig S in S1 Text). Field workers recorded when a macaque would *start*, and when it would *stop*, to groom another individual. We then defined a directed grooming bout as a grooming "start" from $a$ to $b$ that took place at least 30 minutes following the previous grooming "stop" from $a$ to $b$. We obtained a total of 481 directed bouts across 700 directed dyads. We counted bouts within each directed dyad to obtain $y_{[a,b]}$—which we show in Fig 7B–7C (see also Fig T in S1 Text). We established the pedigree from which kin groups $K_{[a]}$ were determined by combining observed birth events, with microsatellite data [87,88]. As in SCM 4, we dichotomised maternal kinship: $Re_{|a,b|} = 1$ if $a$ and $b$ belonged to the same matriline, and $Re_{|a,b|} = 0$ if $a$ and $b$ were born in different matrilines. We modeled each individual's rank $Ra_{[a]}$ from submissive behaviours, using Elo-score point estimates (details in [80]; see also [89,90]).

**Posterior model (empirical data).**   In this section, we describe the posterior model obtained by fitting the empirical data to statistical model 4. To start, we show the marginal posterior distributions for the estimator's fixed effects (Fig 8A). This figure is similar to figures we showed above (e.g., Fig 6B), but differs in two ways. First, the red dots (i.e. the target values) are absent. The "true" structural parameters of the world are, of course, unknown; they are what we are trying to estimate. Second, there is only one posterior distribution per parameter. This is the case because we fitted the model to one empirical data set, instead of several iterations of a simulation.

We observe that 96% of $b_{Re}^{*}$'s posterior distribution is concentrated above 0, with a mean of 0.22 (Fig 8A). It means that the most plausible parameter values for $b_{Re}^{*}$ are positive, given the data and the model. This marginal distribution cannot be intelligibly interpreted by itself in terms of the causal effect of interest nor, therefore, in terms of the species' biology. In fact, the computation of causal effects often requires more than just a slope. They are function of the whole statistical model—an issue we had so far ignored [37].

Generalised linear models (GLM), like statistical model 4, are built-in interaction devices. Contrary to common beliefs, the effect of one parameter on the outcome typically depends on (i.e. interacts with) the value of the other predictors in GLMs, even in the absence of an

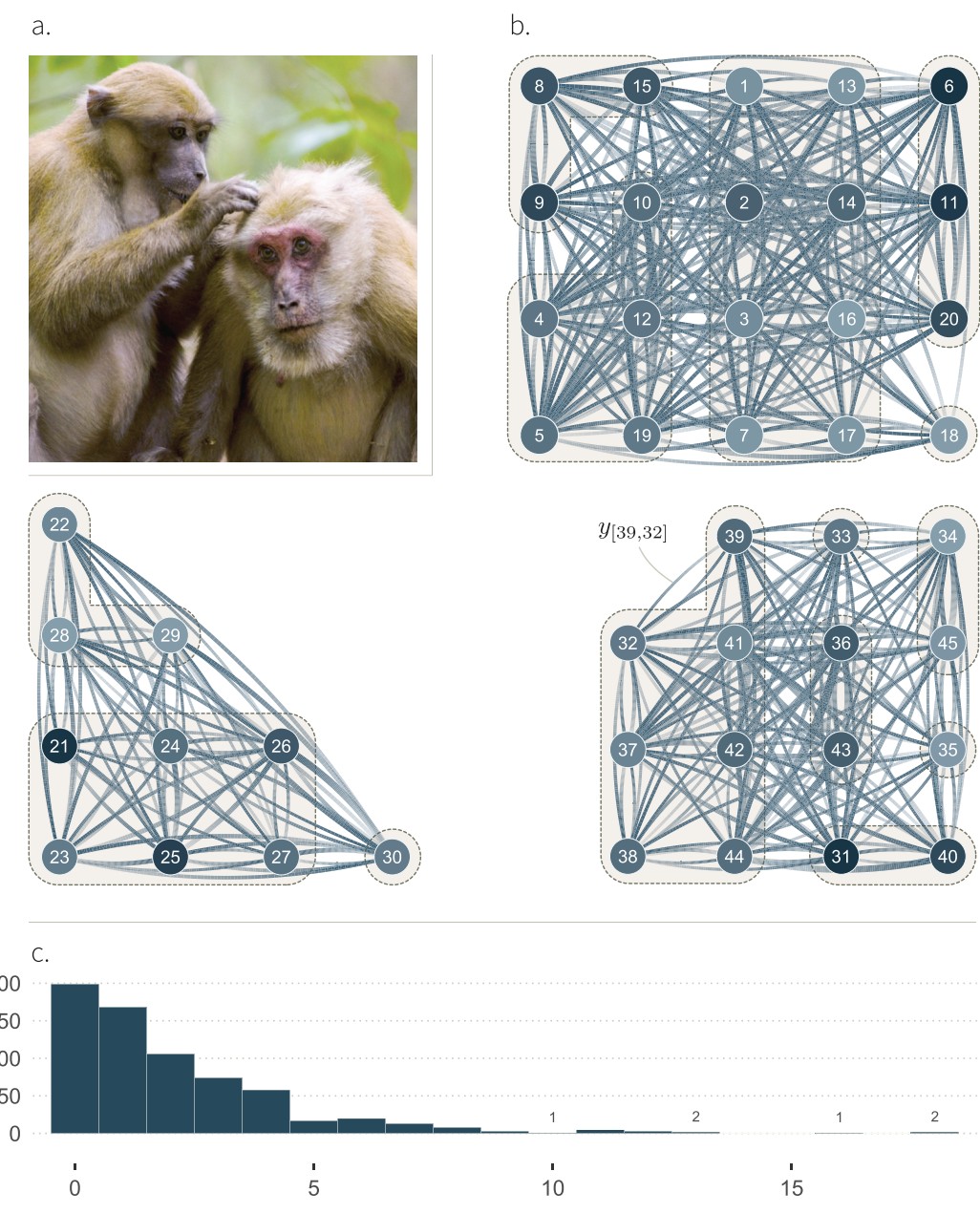

**Fig 7. Empirical system used to illustrate our causal framework: female Assamese macaques. A.** Two Assamese macaques grooming. Photo credit: Kittisak Srithorn. **B.** The graphs show 45 individuals across three groups of respectively 20, 10, and 15 females (we ignore the males). The edges represent the number of observed directed grooming bouts $y_{[a,b]}$ among them. Their width indicates the number of observed interactions: from 0, for no edge, to 18, for the thickest edges. The transparency gradient of the edges corresponds to the direction of the interaction ($y_{[a,b]}$ or $y_{[b,a]}$): the white end of an edge shows the giver, and its darker end shows the receiver. We highlight, as an example, the observed edge $y_{[39,32]}$ (1 interaction) from individual 39 to individual 32. The colour of the node depicts the individual dominance rank: lighter for low ranks (subordinates), and darker for higher ranks (dominants). Kin group are further highlighted by dashed outlines. This network corresponds to the third level of abstraction, in Fig 1. **C.** Distribution of observed directed grooming bouts $y_{[a,b]}$.

a. Marginal posterior distributions

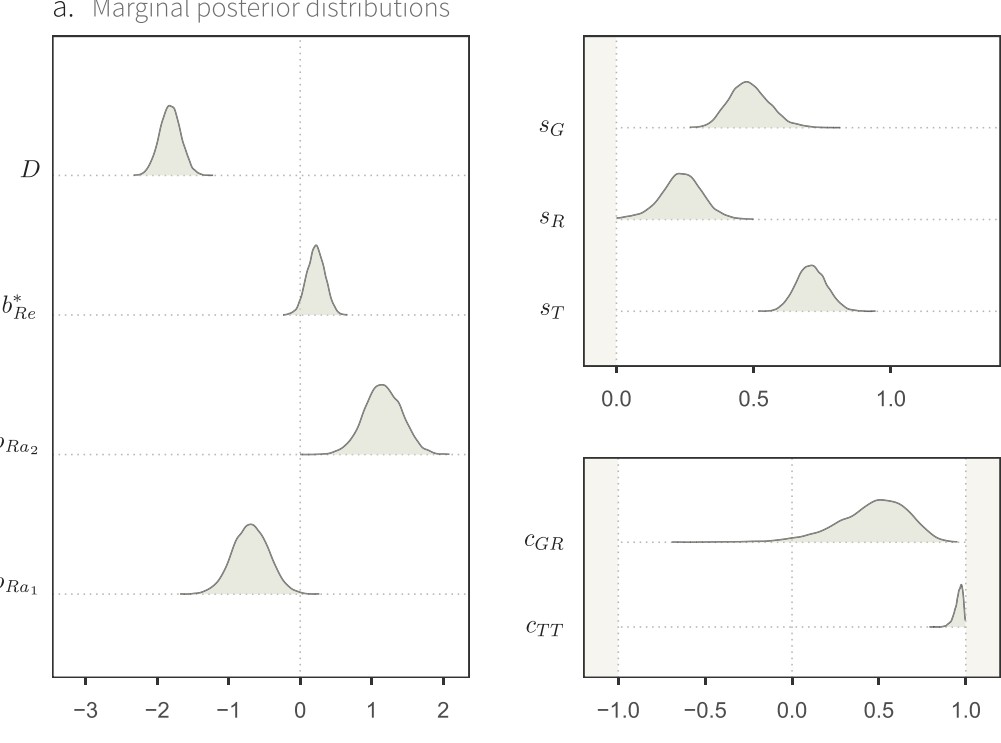

b. Causal effect of kinship on grooming (change in bouts/day) for 3 baseline levels

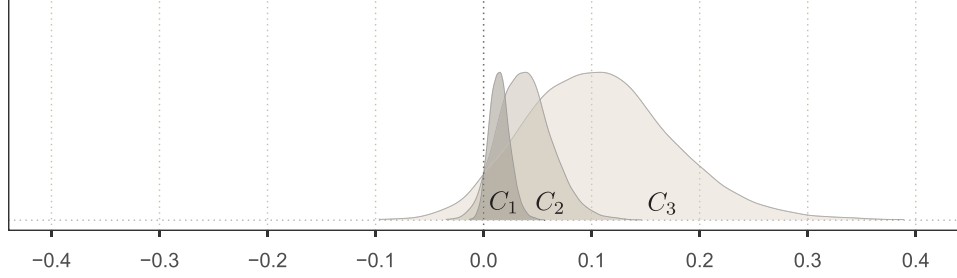

c. Estimated counterfactual grooming rates (bouts/day) for an average dyad

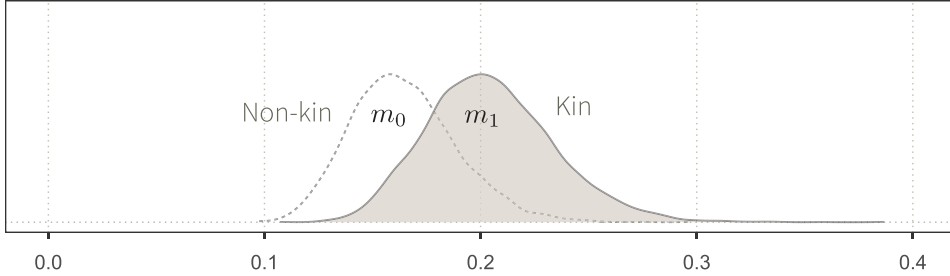

**Fig 8. Posterior model obtained by updating statistical model 4 with empirical data from female macaques. A.**
Marginal posterior distributions for the estimator's fixed effects. **B.** Conditional average relatedness effect $C_j$ on inferred
grooming rate $m_{[a,b]}$ (posterior mean contrast). We show this effect for three hypothetical dyads, respectively varying in
their "baseline" level of grooming rate $\hat{\psi}_j$ (see main text)—from darkest to lightest: low-, medium-, and high-baseline.
**C.** Counterfactual rates $m_0$ and $m_1$ for a hypothetical average directed dyad, where the two individuals occupy the same
rank in the dominance hierarchy. The difference between the two distributions represents the causal effect of genetic
relatedness for that dyad.

interaction term [91,92]. Imagine a simple GLM with a slope of value 2 quantifying the causal effect of a binary variable $X \in \{0, 1\}$ on a rate $\mu$, for $i$ observations:

$$\mu_i = \exp(\alpha + 2 \cdot X_i)$$

Now suppose that the intercept $\alpha$ can take two different values: 0 and 1. In the case where $\alpha = 0$, the causal effect of $X$ is the following: $\exp(0 + 2 \cdot 1) \simeq 7.4$, minus $\exp(0 + 2 \cdot 0) = 1$, which is equal to 6.4. If, however, $\alpha = 1$, then the average effect is $\exp(1 + 2 \cdot 1) \simeq 20$, minus $\exp(1 + 2 \cdot 0) \simeq 2.7$, equal to $\simeq 17.4$. So, the same slope implies two very different effects: 6.4 and 17.4. With an exponential "inverse-link" function, causal effects on the outcome scale are multiplicative: they are larger when the baseline value—here, the value of $\alpha$—increases.

The same principle applies to statistical model 4, where the effect of the slope $b_{Re}^*$ on the rate $m_{[a,b]}$ differs across dyads—i.e. a larger effect for dyads with a higher "baseline" level of grooming interactions. We define these differences in baseline using a parameter $\psi_{[a,b]}$ ("psi"):

$$\psi_{[a,b]} = D + G_{[a]} + R_{[b]} + T_{[a,b]}$$

Where the distribution of $\psi_{[a,b]}$ captures the variation across dyads that is caused by the unobserved (or exogenous) network structuring features—like age, social group differences, "personality", or friendship—as captured by Eqs 4.2.2–4.2.3. We show the distribution of $\psi_{[a,b]}$ in Fig Z in S1 Text. We roughly summarise it with three values: $\hat{\psi}_j \in \{-2.8, -1.9, -0.9\}$, which respectively represent dyads with a low-, medium-, and high-baseline levels of grooming rate Fig 8B. Equivalently, we can think of $\hat{\psi}_1$, $\hat{\psi}_2$, and $\hat{\psi}_3$ as standing for negative, null, and positive deviations from the global intercept $D$. For instance, a directed dyad with baseline $\hat{\psi}_1 = -2.8$ implies that unobserved factors are causing $a$ to groom $b$ at a low rate, compared to the other dyads in the population. We then compute the causal effect of genetic relatedness $C$, for each of these three levels $\hat{\psi}_j$ (Fig 8B):

$$C_j^{(n)} = \exp(\hat{\psi}_j + 1 \cdot b_{Re}^{(n)}) - \exp(\hat{\psi}_j + 0 \cdot b_{Re}^{(n)}),$$

That is, we obtained a posterior distribution for the contrasts of interest $C_j$ from manipulating the Markov Chain Monte Carlo (MCMC) samples of the posterior model: each contrast $C_j$ is composed of 8000 posterior samples $n$: $C_j^{(1)}, \ldots, C_j^{(8000)}$. This causal effect is sometimes called *Conditional Average Treatment effect* (CATE), because its value depends on—is *conditional* on—the value of the other predictors. Our model shows an expected increase of, respectively, 0.01, 0.04 and 0.11 grooming bouts per day due to genetic relatedness, for low-, medium-, and high-baseline interaction levels, respectively (that is, about an order of magnitude change from low- to high-baseline). These values correspond to the posterior CATE means. Note, however, that the estimates' distributions are rather broad. Albeit unlikely, small negative effects are still plausible from the estimator's perspective.

On the last panel (Fig 8C), we show our causal estimate in a different way. We compare $m_{Re=1}$ ($m_1$) and $m_{Re=0}$ ($m_0$), where:

$$m_1^{(n)} = \exp(D^{(n)} + 1 \cdot b_{Re}^{(n)})$$
$$m_0^{(n)} = \exp(D^{(n)} + 0 \cdot b_{Re}^{(n)}).$$

Again, the $n$ index indicates 8000 draws from the posterior distribution. These two estimates represent two counterfactual outcomes for a hypothetical average dyad. That is, they

correspond to estimated grooming rates of a dyad, if the two individuals were unrelated (left) or if they were kin (right), given that everything else remains identical: $G_{[a]}$, $R_{[b]}$, and $T_{[a,b]}$, and the difference in rank between them are set to zero. Comparing the two distributions allows us to visualise the relative effect of genetic relatedness. The posterior estimates have a mean of 0.16 and 0.20 grooming bouts per day, respectively. 0.20 is equal to 1.24 times 0.16; so relatedness is estimated to cause a relative increase of about 24% in grooming rate. Note that the difference between the two distributions roughly corresponds to $C_2$ in Fig 8B.

**Summary and discussion.**  We do not wish to further discuss these results in terms of the female Assamese macaques's biology—it would go beyond the scope of the manuscript. We also do not, in any way, argue that our estimator is optimal and that it captures all necessary factors at play. Instead, we built this example to illustrate two general points. Our first point regards the logical implications of causal assumptions for the development of an estimator. By examining our DAG, we showed that it was *necessary* to block a backdoor path (by conditioning on dominance rank) to identify our estimand. Not doing so resulted in a consistently biased results. Furthermore, encoding the DAG as a SCM has made visible that we needed to account for the dyadic *difference* in rank between individuals—not only for the individuals' ranks—, and to allow for this effect to possibly change depending on whether the difference was positive or negative. These considerations arose from relatively consensual biological knowledge about the social system (e.g., see [8,83]). Yet, they were not obvious from intuition alone.

Translating domain expertise into formal assumptions allowed us to *logically* connect causal assumptions and their implications for the estimator structure, thereby providing a licence for this estimator. Failing to establish this connection places an analysis on loose grounds, for it becomes impossible to know under which conditions it can work, even in theory. For instance, most empirical studies of the effect of maternal kinship on affiliation in Cercopithecinae have not considered the biasing path of rank (e.g., [93–95]). It is, however, hard to know if the confounder was simply ignored, or if instead, the results hold under other assumptions—in which case, we do not know which ones.

Secondly, we discussed why computing causal estimates on the outcome scale usually involves not only a slope, but the rest of the posterior model too. We notably showed that in the population under study, the effect substantially changed across dyads depending on their baseline interaction level. These considerations are particularly important because they determine how to interpret evidence for causal effects in light of the species' biology.

We refer the interested readers to supplementary sections where we explore other aspects of this study. We show MCMC diagnostics in Sect H.2 and posterior predictive checks in Sect H.3 of S1 Text. We also compute the effect of rank—which, like that of relatedness, cannot be interpreted from the marginal posterior distributions alone—in Sect H.5 of S1 Text. Finally, we introduce simulation study 4', where group sizes also affects interaction rates through a novel parameter $\delta$: a group-level counterpart of $\gamma$, $\rho$, and $\tau$ (Sect I in S1 Text).

## General discussion

Behavioural ecologists study the phenotypic and ecological factors that affect the structure of animal societies. This body of research is grounded in theoretical models, whether verbal or formal, that provide potential *causal explanations* for patterns of behavioural variation. These models have highlighted the roles of processes at the individual level (e.g., age, reproductive state; [8,79]), at the dyad level (e.g., dominance, kinship, homophily, friendship; [8,15,96]), and at the supra-dyadic level that can shape the social relationships between individuals (first level of abstraction, Fig 1), and thus, the social network structure emerging from

it. Apace with this, there has been growing interest in empirically studying these processes in wild and captive populations. However, the connection between empirical results and theoretical models is often unclear. Many empirical studies are causally ambiguous, and theoretical assumptions specifying how variables cause one another in the system are rarely spelled out [61,63,69]. In addition, social network data analyses rarely integrate the measurement process, e.g., by assuming that the observed social interactions perfectly capture the underlying interaction network (Fig 1). Statistical models that are not built in light of these biological and sampling processes can produce noisy and biased estimates, yielding incorrect results—as we outlined in problems I–IV.

# Box 1

**A simple workflow for causal inference in animal social networks**

1. *Define the estimand*, i.e. the theoretical quantity that the analysis is designed to estimate [66].

2. *Draw a DAG*, specifying qualitative assumptions about the biological and sampling processes that generate the data. An adjustment set can be derived from this DAG, for instance using the backdoor criterion (see [67]). Note that certain confounding paths might be automatically blocked by the varying-effects of the Social Relations Model (e.g., Sect I in S1 Text).

3. *Translate the DAG as a SCM*, encoding plausible quantitative assumptions about the data generating process [37,67]. The SCM should ideally encode a target value for the estimand, which can be recovered by an estimator. We recommend exploring the parameter space to better understand how the simulation behaves.

4. *Build an estimator* based on the adjustment set derived from the DAG, and following the functional assumptions encoded in the SCM. The basic architecture of the Social Relations Model is a reasonable starting point for many animal social network models [23,38,72].

5. *Validate that the estimator can recover the estimand*, by fitting the synthetic data generated by the SCM to the estimator. To better understand how the estimator works, we recommend inspecting not only the marginal posterior distribution of the parameter(s) of interest, but also, plotting other parameters of the posterior distribution (e.g., $s_G$, $c_{TT}$, $G_{[a]}$, $m_{[a,b]}$), as well as computing causal effects on the outcome scale.

6. *Loop back to 1.*, adding one layer of complexity. The new layers may involve any of the step above, like refining the estimand, adding a variable in the DAG, or encoding a more realistic functional relationship. After repeating this cycle, if you believe that you have translated your domain expertise into assumptions that reasonably approximate the data generating process, and that your estimand is recoverable from the estimator, then go to the next step.

7. *Fit the empirical data to the estimator*, and compute causal effects from the joint posterior distribution.

Here, we have advanced a general framework for studying the causes of animal social network structure that allows empiricists to translate their theoretical domain expertise into an

analytical strategy. *Within* each simulation study, we showcased how researchers may, first, define causal assumptions at various levels (e.g., individual, dyad, group), and second, how they can validate an estimator for the effect of interest given these assumptions. We highlighted how these models naturally deal with (I) the gap between the interaction network and the observed network (discussed in simulation 1), (II) variation in sampling effort across individuals and dyads (sim. 3), (III) counfounders (sim. 4) and (IV) unobserved network causes (sim. 1-3). *Across* the four studies, we built models layer by layer, and illustrated an iterative workflow for causal inference in social networks, which we summarise in Box 1. In doing so, we provided empiricists with reproducible analytical tools that they can build upon to specify causal and statistical models for their own study system (see our GitHub repository).

## Practical application

The steps in Box 1 represent the core components of a full workflow for causal inference in animal social networks. Yet, they omit certain complexities. First of all, there often exist several plausible causal models that are compatible with a researcher's domain expertise. In this case, it can be useful to run several analyses in parallel and to compare their outputs—that is, the list of Box 1 bifurcates into forking paths. Fitting empirical data to estimators of intermediate complexity can also be insightful to check whether we observe the conditional dependencies that are expected from the DAG [67]. Furthermore, predictive tools (e.g., Information Criteria, Cross Validation) can be useful to compare causally consistent estimators that differ in their parametric specifications (see [97]). Finally, we did not highlight common aspects of a Bayesian workflow in the main text, like prior and posterior predictive checks, or MCMC diagnostics (see [37,98]). These procedures are, however, important. We showed how they can be conducted for social network models in S1 Text (e.g., Sects H.2–H.3).

Finally, although the framework presented in Box 1 applies to empirical research across animal social network analysis, the modelling details may require substantial changes—this is notably the case when modelling association data, as measured by spatio-temporal co-occurrences. Association data are not truly dyadic, and much of the observed variation is often unrelated to the underlying social relationships [18]. Thus, we believe that the models presented in this manuscript may often be poor approximations of the processes generating them. The literature on *hypergraph models* [99–102] represents a promising avenue for the analysis of association data, though much work is still needed for accessible and principled causal inference in this context.

## Future research avenues

**Measurement model.** Conducting causal inference in social networks forces us to carefully examine the mapping between latent constructs of interest and data. In this manuscript, we focused on two kinds of unobserved quantities: true interaction rates, and causal effects. However, empirical research often involves additional latent variables, whose true values are uncertain or unknown. Let us return to the macaque example. We notice that, in addition to the unobserved causal effect on the unobserved rate $m_{[a,b]}$, the very variable whose effect we studied—genetic relatedness—was not directly measured either. In the study, we *approximated* it as a binary variable using pedigree and microsatellite data, thereby discarding the continuous variation and uncertainty in the true genetic relatedness among individuals. Furthermore, we were interested in studying its effect on *affiliation*, a theoretical construct that we approximated with grooming rate (Fig 1). In doing so, we ignored the rich set of behavioural interactions that compose affiliation in this species. Finally, by using Elo-rating point estimates for dominance rank, we treated this variable as if it was directly measured.

It is important to realise that whenever a variable is unobserved, its relationship with observed variables depends on theoretical assumptions, whether explicit or implicit. In this case, we implicitly assumed that the proxies captured the latent variables without any uncertainty or error, thereby making a *leap* between levels of abstractions. This is far from optimal, for it can lead to inefficient and inaccurate estimators.

Moving forward, we argue that such latent variables should be modelled explicitly when possible, as part of a *measurement model*: i.e. a joint set of assumptions defining the connection between observed and unobserved variables. Doing this is natural with Bayesian models, where latent variables are assigned a posterior distribution like any other parameter. In the empirical study, genetic relatedness could for instance be *estimated* alongside the other components of the model [82]. Similarly, dominance relationships and affiliation may be modelled as latent variables in multiplex network models [22–24,45,103,104]. In this regard, statistical model 4 can be considered an intermediate step towards more realistic estimators. These considerations further apply to traits like age, whose exact value is sometimes unknown, but which can be modelled given a plausible interval; e.g., if an individual is known to be born between two population surveys [37]. Furthermore, we believe that explicit measurement models could clarify current debates in fields like animal personality research, where the link between the latent objects of study (personality, behavioural syndrome) and the observed variables (behavioural measures) is conceptually and analytically challenging [105–107].

**Dynamical drivers of social network structure.** Sometimes, the latent causes of network structure cannot be modelled directly, but instead, can be inferred from the multilevel structure of the statistical model. This is the case for dynamic processes like reciprocity, a plausibly important driver of social network structure across social species. For instance, the Social Relations Model's $c_{TT}$ parameter may be interpreted as quantifying the extent of *dyadic reciprocity* after (i) positing a specific mechanism for reciprocity in a SCM or a more fine-grained agent based model (e.g., [37,108]), (ii) under the assumption that all confounding paths between $T_{[a,b]}$ and $T_{[b,a]}$ have been blocked, and (iii) after verifying that given (i) and (ii), the pattern of interest could be detected with $c_{TT}$. The same goes for $c_{GR}$ as a tool to quantify generalised-reciprocity. These considerations not only apply to animal social network analysis, but also to several disciplines in the Social and Behavioural Sciences—e.g., Psychology or Anthropology—where the Social Relations Model's parameters are interpreted in terms of latent processes [23,46,109–111]. In any case, we wish to insist that such inferences require extreme care. As we saw earlier, most causal paths flowing through $\gamma$, $\rho$, and $\tau$ end up in $c_{TT}$ and/or $c_{GR}$. Therefore, these paths would need to be all blocked for $c_{TT}$ and $c_{GR}$ to be interpret as meaningful signals of reciprocity. As always, the causal evidence will only be as strong as the causal assumptions are plausible. But remember, imperfect causal assumptions are still better than no explicit assumptions at all [112].

An area where the causal tools presented here may be particularly useful is to study the drivers and consequences of social network structure in the context of longitudinal data analysis [73,113,114]. In our studies, we always assumed stationary systems, where time did not matter. However, many network structuring processes are intrinsically dynamic. For instance, an individual's gregariousness might affect its health, reproductive success, and survival [115–117]. These outcomes, in turn, can shape social relationships and the pool of individuals in the social network, thereby forming potential loops of reciprocal causation between social network structure and other phenotypic traits [118,119]. Another way for social network structure to affect itself over time regards triadic closure [96,120]. Consider three individuals, *a*, *b*, and *c*. If *a* and *b* are both connected to *c* at time *t*, they might become more likely to form

a connection $a$–$b$ at time $t+1$, *because* of their shared relationship with $c$ ("*friends of friends become friends*").

Social network structuring processes, along with the measurement procedure to capture them, are intrinsically causal. They all pose the inferential challenges that we have outlined throughout the paper: when not integrated into an analysis, they can lead to inferences that are simply wrong. In this context, we believe that establishing a logical connection between theory and data, using transparently justified estimators, is crucial. We hope that our proposed tools and workflow will inspire future empirical research in this effort.

## Ethics approval

The empirical data used in this manuscript were collected non-invasively, using protocols that adhere to the Association for the Study of Animal Behaviour (ASAB) guidelines for the Use of Animals in Research. The study was further authorised by the Department of National Parks, Wildlife and Plant Conservation of Thailand, and the National Research Council of Thailand with a benefit-sharing agreement (permit number: 0002/4137).

## Figures

The figures in this manuscript were generated using *R* (version 4.2.1), LaTeX (TikZ package), *Adobe Illustrator CC 2015*, or a combination of them. In R, we used the following packages: *ggplot2* [121], *tidybayes* [122], *patchwork* [123], *ggraph* [124], and *igraph* [125].

## Supporting information

**S1 Text. Supplementary materials.** Complementary figures, complete definitions of all statistical models, prior predictive simulations, posterior diagnostics, variations on simulation studies 2–4, and alternative parameterisations of several models.
(PDF)

## Acknowledgments

We thank Alice Hill, Ana Lucia Arbaiza Bayona, and Shivani for their input on earlier versions of the manuscript. We acknowledge support by the Open Access Publication Funds of the Göttingen University.

## Author contributions

**Conceptualization:** Ben Kawam, Julia Ostner, Richard McElreath, Oliver Schülke, Daniel Redhead.

**Formal analysis:** Ben Kawam.

**Funding acquisition:** Julia Ostner, Oliver Schülke.

**Methodology:** Ben Kawam, Richard McElreath, Daniel Redhead.

**Project administration:** Julia Ostner, Oliver Schülke.

**Resources:** Julia Ostner, Oliver Schülke.

**Software:** Ben Kawam.

**Supervision:** Julia Ostner, Richard McElreath, Oliver Schülke, Daniel Redhead.

**Validation:** Ben Kawam.

**Visualization:** Ben Kawam.

**Writing – original draft:** Ben Kawam, Daniel Redhead.

**Writing – review & editing:** Ben Kawam, Julia Ostner, Richard McElreath, Oliver Schülke, Daniel Redhead.

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
