## [Decision Letter · Decision Letter 0]

1 Apr 2025

PCOMPBIOL-D-24-02125

A causal framework for the drivers of animal social network structure

PLOS Computational Biology

Dear Dr. Kawam,

Thank you for submitting your manuscript to PLOS Computational Biology. After careful consideration, we feel that it has merit but does not fully meet PLOS Computational Biology's publication criteria as it currently stands. Therefore, we invite you to submit a revised version of the manuscript that addresses the points raised during the review process.

Please submit your revised manuscript within 30 days Jun 01 2025 11:59PM. If you will need more time than this to complete your revisions, please reply to this message or contact the journal office at ploscompbiol@plos.org. Please include the following items when submitting your revised manuscript:

We look forward to receiving your revised manuscript.

Kind regards,

Christian Hilbe

Academic Editor

PLOS Computational Biology

Zhaolei Zhang

Section Editor

PLOS Computational Biology

**Additional Editor Comments:**

Thanks for submitting this paper. It has been now evaluated by two experts. They both agree that the paper is an extremely well-written manual on how to do causal inference with certain types of animal network data, and I concur. The reviewers provide some very constructive suggestions for how to further improve the manuscript. Once these suggestions are incorporated, the manuscript will be a great fit for PLoS Computational Biology.

**Journal Requirements:**

4) We notice that your supplementary Figures are included in the manuscript file. Please remove them and upload them with the file type 'Supporting Information'. Please ensure that each Supporting Information file has a legend listed in the manuscript after the references list.

Potential Copyright Issues:

i) Please confirm (a) that you are the photographer of Figure 7A, or (b) provide written permission from the photographer to publish the photo(s) under our CC BY 4.0 license.

6) Please amend your detailed Financial Disclosure statement. This is published with the article. It must therefore be completed in full sentences and contain the exact wording you wish to be published. State the initials, alongside each funding source, of each author to receive each grant. For example: "This work was supported by the National Institutes of Health (####### to AM; ###### to CJ) and the National Science Foundation (###### to AM).".

**Reviewers' comments:**

Reviewer's Responses to Questions

**Comments to the Authors:**

Reviewer #1: This manuscript describes a causal inference methodology to reconstruct animal social networks from data.

The main purpose of the manuscript seems to be to present to the community of behavioural ecology how causal inference can be performed on observational data, rather than to develop entirely new methodology.

The manuscript is exceptionally well written and carefully didactic, without being burdensome. It elucidates quite clearly the distinction between abstraction (model) and data, which is very often conflated in ecology and network science at large. At the center is the role of causal inference via directed acyclic graphs (DAGs) and structural causal models (SCMs): making (inescapable) causal assumptions explicit, so that causal conclusions can be made coherently from observational data. In the absence of interventions or controlled randomized experiments, causal conclusions can only be extracted if conditioned on a priori causal assumptions. The authors explain how these can be formally expressed via DAGs, and incorporated into generative statistical models via SCMs, whose parameters can be inferred from data using standard Bayesian methodology. The whole approach provides a comprehensive, principled, and open-ended pipeline that is powerful, and should be more widely adopted than it currently is.

The authors employ this framework to the so-called "social relations model", a network "measurement" model that is applied to empirical pairwise iteractions between macaques, and incorporates relevant features such kinship and ranks, besides node and dyad-level propensities.

Overall, the case is made convincingly. My only concern is that someone already familiar with Bayesian inference of hierarchical models, but less so with causal models, might wonder what is the added value of incorporating a formal causal framework for such problems. For example, the difference between the rank-stratified model of Eq. 4.2.3 and its variant without stratification might just be interpreted as the model being mispecified or not. In other words, following the workflow of Box 1 (page 39), why not go from step 1 directly to step 4 with a statistical generative model? The answer is presumably that a statistical model does not necessarily encode the causal assumptions unambiguously. But in simple examples (like many considered in the manuscript) they are fairly evident from the generative model. I believe this could be expanded more.

Some more minor observations:

1. When referring to the literature of Bayesian network reconstruction, the seminal work of Butts should be mentioned: https://doi.org/10.1016/S0378-8733(02)00038-2

2. Furthermore, there's a large volume of work on Bayesian network inference using structured priors that is immediately relevant and should also be mentioned, e.g. https://doi.org/10.1073/pnas.0908366106, https://dx.doi.org/10.1103/PhysRevX.8.041011, https://dx.doi.org/10.1103/PhysRevX.12.011004

3. In page 13, line 322, the authors mention that causal effects on count data are “unavoidably” multiplicative, instead of additive. However the model formulation of Eq. 1.1.4 is a choice... It could have been linear, provided *γ*, *ρ*, and *τ* were restricted to non-negative values (also a valid modeling choice).

4. With respect to Eqs. 1.1.4 and 1.2.2 and so on, the authors should discuss a potential identifiability problem with the values *γ*, *ρ*, and *τ* and G, R, T, and so on, since it's possible to redistribute their values while keeping their sum fixed, which would preserve the data likelihood. This symmetry is "broken" by the priors on each parameter, but the non-identifiably will still manifest itself in an (artificial) dispersion in the posterior distribution. In some cases like in the degree-corrected SBM this is resolved by normalizing the node parameters (without loss of generality) but this requires a change in the priors.

Reviewer #2: The manuscript “A casual framework for the drivers of animal social network structure” presents and acts as a tutorial for a way of understanding and studying animal social networks. It presents both the framework itself, but also a guide about how to apply it- as well as incidentally along the way acting as a excellent tutorial in casual thinking in general, and Bayesian analysis in general.

The framework is very exciting, very interesting and will make an important positive impact in the field. The manuscript is exceptionally well-structured, well-written and clear- in the main text, in the supplementaries and in the GitHub repository. It well justifies it’s length. I congratulate the authors on a brilliant paper, and look forward to seeing it published.

I have one general comment and a series of minor comments, but they are all advisory rather than important, and I would be very happy for the authors to use their discretion about which comments they find useful and which they don’t.

My more general comment is, that it would be valuable either in the Introduction section(s) and/or Discussion to highlight how this framework applies or does not apply to other kinds of data common in animal social network analysis. I’m thinking explicitly of non-directional data such as association/gambit-of-the-group data. Both of which are commonly collected and analysed (including in many of the references in the first few sections). The framework presented here is explicitly designed for directional data. I assume the framework could be adapted to deal with non-directional data, but in it’s current form the framework is designed for directional data. I don’t think it needs to be extended to this kind of data here (there is plenty in this manuscript already!)- but an explicit acknowledgement of this might help avoid confusion, and guide readers to the right framework and tools for the data/questions they have.

I also have a series of minor comments. These were notes I made as I read at points where things were not completely clear to me on first reading. My points are all places where I did not follow and had to think and read back, or questions that occurred to me as I read. They are completely advisory, but I include them because if they are points of small confusion to me they might also be to other readers. I am happy for these comments to be ignored at will- they could just be me.

Line 17 “M.J. Silk” There are a few points through the manuscript (e.g. line 26 in this paragraph as well) where the references seem to add initials or first names to some authors. Could be worth a careful read through to check.

Line 23 It might be worth acknowledging that some studies are not interested in the first level of construct e.g. studies of disease spread the realised rate of some behaviour is more important than the social bonds it represents.

Line 81 they also sometimes discard individuals with insufficient data collected….

Line 82-83 Though attempts have been made, perhaps most influentially: Whitehead, Hal. ‘Precision and Power in the Analysis of Social Structure Using Associations’. Animal Behaviour 75 (2008): 1093–99. https://doi.org/10.1016/j.anbehav.2007.08.022.

Line 257-258 It would help DAG amateurs (such as myself) to link “casual” and “non-causal” paths here to the direction of the arrows in the DAG. My reading is the casual is with the arrows, non-casual involves going backwards along an arrow.

Line 262-263 Similarly I read directed here as “with the arrows”

Line 339-341 Might be worth a reminder of what the authors mean about “aggregated indices” could be as simple as just adding an “(e.g. SRI, DSI etc)”.

Footnote 5. The term “gregariousness” is the word I associate with popularity in social network analyses.

Line 515-523 Here you refer to the models as “well-specified” and “non-adjusted” but in the figure it is “adjusted” and “non-adjusted”

Line 524 If this is intended to be a general introduction a brief unpacking of the word “conditioned” here might help those unfamiliar with the DAG language.

Line 556-561 Why do we expect Sampling effort to effect tau but not rho and gamma? I think I have an answer but it might help readers to have it stated explicitly.

Just below equation 3.2.3 (the line numbers seem to pause for a while here?). Would it be correct or useful to describe T-hat as “realised social bond” or something like it?

Line 655 This line made me expect to see rank modelled as linked to rho- i.e. high ranked individuals are more popular than low ranking individuals. It would be interesting to hear why this was not the case.

Line 666 Define/remind what an “open path” means

Line 758 This is an important point and I worry that the word “interacts” might muddy the water, given that a lot of people have an explicit idea of what an interaction (which is not what is being discussed here).

P36 above the first psi equation (line numbers paused again). An explicit pointer back to equations 4.2.2 and 4.2.3 might be helpful, I found myself flicking back.

Line 764 “middle contrast” = C2 ?

Line 860 I don’t quite follow what you mean by:“that is, the list branches out”.

**Have the authors made all data and (if applicable) computational code underlying the findings in their manuscript fully available?**

Reviewer #1: Yes

Reviewer #2: Yes

PLOS authors have the option to publish the peer review history of their article (what does this mean?). If published, this will include your full peer review and any attached files.

Reviewer #1: No

Reviewer #2: **Yes: **Samuel Ellis

**Figure resubmission:**
---

## [Decision Letter · Decision Letter 1]

10 Jul 2025

PCOMPBIOL-D-24-02125R1

A causal framework for the drivers of animal social network structure

PLOS Computational Biology

Dear Dr. Kawam,

Thank you for submitting your manuscript to PLOS Computational Biology. After careful consideration, we feel that it has merit but does not fully meet PLOS Computational Biology's publication criteria as it currently stands. Therefore, we invite you to submit a revised version of the manuscript that addresses the points raised during the review process.

Please submit your revised manuscript within 30 days Sep 09 2025 11:59PM. If you will need more time than this to complete your revisions, please reply to this message or contact the journal office at ploscompbiol@plos.org. Please include the following items when submitting your revised manuscript:

We look forward to receiving your revised manuscript.

Kind regards,

Christian Hilbe

Academic Editor

PLOS Computational Biology

Zhaolei Zhang

Section Editor

PLOS Computational Biology

**Additional Editor Comments:**

The revised manuscript has been sent the two original reviewers, and both of them are very pleased with the revised version. So from a scientific point of view, the paper is ready to be published.

Now, as it stands, the paper seems to be formatted in a way that is not fully compatible with the usual PLoS Computational Biology formatting (e.g., the glossary items in the margins of the paper; if these are important, perhaps you can collect them in a single display item). Ideally, please provide us with a version that follows as closely the general guidelines as possible (however, we might be willing to be a little bit more flexible in your case).

Also, it might well be that the best format for your paper is not the usual 'article' type, but rather a 'methods submission': https://journals.plos.org/ploscompbiol/s/submission-guidelines#loc-methods-submissions

However, I would leave it to the authors' judgment which article type best serves their needs.

**Journal Requirements:**

1) Please upload all main figures as separate Figure files in .tif or .eps format. For more information about how to convert and format your figure files please see our guidelines: 

2)  Please ensure that the funders and grant numbers match between the Financial Disclosure field and the Funding Information tab in your submission form. Note that the funders must be provided in the same order in both places as well.  

**Reviewers' comments:**

Reviewer's Responses to Questions

**Comments to the Authors:**

Reviewer #1: The authors have adequately addressed my remarks. The current version of the manuscript is suitable for publication.

Reviewer #2: I'm happy to see this paper again, and happy with the response to my comments. I thank the reviewers for engaging so thoroughly with them. I have nothing further to add.

**Have the authors made all data and (if applicable) computational code underlying the findings in their manuscript fully available?**

Reviewer #1: Yes

Reviewer #2: None

PLOS authors have the option to publish the peer review history of their article (what does this mean?). If published, this will include your full peer review and any attached files.

Reviewer #1: No

Reviewer #2: No

**Figure resubmission:**
---

## [Editor Report · Decision Letter 2]

26 Jul 2025

Dear Mr. Kawam,

We are pleased to inform you that your manuscript 'A causal framework for the drivers of animal social network structure' has been provisionally accepted for publication in PLOS Computational Biology.

Best regards,

Christian Hilbe

Academic Editor

PLOS Computational Biology

Zhaolei Zhang

Section Editor

PLOS Computational Biology

---

## [Editor Report · Acceptance letter]

PCOMPBIOL-D-24-02125R2

A causal framework for the drivers of animal social network structure

Dear Dr Kawam,

I am pleased to inform you that your manuscript has been formally accepted for publication in PLOS Computational Biology. Your manuscript is now with our production department and you will be notified of the publication date in due course.

With kind regards,

Lilla Horvath
